# OoD-Control: Out-of-Distribution Generalization for Adaptive UAV Flight Control

## Abstract

Data-driven control methods have demonstrated precise and agile control of Unmanned Aerial Vehicles (UAVs) over turbulence environments. However, they are relatively weak at taming the *out-of-distribution* (OoD) data, i.e., encountering the generalization problem when faced with unknown environments with different data distributions from the training set. Many studies have designed algorithms to reduce the impact of the OoD problem, a common but tricky problem in machine learning. To tackle the OoD generalization problem in control, we propose a theoretically guaranteed approach: *OoD-Control*. We provide proof that for any perturbation within some range on the states, the control error can be upper bounded by a constant. In this paper, we present our OoD-Control generalization algorithm for online adaptive flight control and execute it in two instances. Experiments show that systems trained by the proposed OoD-Control algorithm perform better in quite different environments from training. And the control method is extensible and pervasively applicable and can be applied to different dynamical models. OoD-Control is validated on UAV dynamic models, and we find it performs state-of-the-art in positioning stability and trajectory tracking problems.

## 1 Introduction

UAVs have gained considerable attention and are widely used for various purposes because of their high manoeuvrability and flexibility. For example, quadrotors are widely deployed for inspection, reconnaissance, and rescue. As control strategies evolve, novel scenarios for UAVs, such as aerial grasping, transporting, and bridge inspection (Ruggiero et al., 2018), require more precise trajectory tracking. Especially in the outdoor environment, unpredictable and changing wind field conditions pose substantial challenges to the stability of UAVs. Rotor blades are affected by induced airflow caused by the wind, which creates complex and non-stationary aerodynamic interactions (see Appendix B.6.3). From security and policy perspectives, demonstrating that UAVs can operate safely and reliably in unpredictable environments with various distributions is an essential requirement. It is also the premise for future medical robots, autonomous cars, and manned aerial vehicles to be widely accepted.

Many areas have benefited from data-driven approaches. However, they are susceptible to performance degradation after generalization. And the majority of deep learning algorithms heavily rely on the I.I.D assumption for data, which is generally violated in practice due to domain generalization (Zhou et al., 2022). Nevertheless, neural networks may lose their robustness when confronted with OoD data. Many cases of failure in DNN originate from shortcut learning in the learning process (Geirhos et al., 2020). The damage to the UAV is undoubtedly considerable if the UAV cannot adjust to the changing environment, i.e., it is unstable or even crashes in an OoD situation. One significant objective of this paper is to propose a control algorithm to enable UAVs to maintain accurate control even in the case of environment domain shifts.

**Our Contributions.** UAVs interact with the changing environment, resulting in complex environment-dependent uncertain aerodynamics, called *unknown dynamics*, that are tricky to model and significantly impact precise control. Previous data-driven controllers attempt to solve the problem by estimating the unknown dynamics, while the estimation accuracy and the performance of the controllers are limited by the environment domain shifts in tests. This paper presents a methodology

for adaptive flight control problems, focusing on enabling UAVs to fly under unknown environments. Compared with previous works, our proposed OoD-Control algorithm can provide performance guarantees under domain shifts of the environment distribution. Compared with previous state-of-the-art work (Shi et al., 2021), the proposed OoD-Control method does not require strong assumptions, for example, e-ISS stability and a fully actuated system. Additionally, our algorithm has a greater capacity for generalization. For different distributions of the environment, we show theoretically that the bound on the prediction error of the unknown dynamics remains constant over a certain range of perturbations. Besides, simulated results under challenging aerodynamic conditions indicate that the OoD-Control algorithm achieves better control performance than the SOTA deep learning algorithms.

## 2 RELATED WORK

### 2.1 FLIGHT CONTROL ALGORITHMS

UAVs have found broad applicability in a variety of fields and have attracted the attention of several researchers. Many published studies describe the significance and efficiency of flight control algorithms, including PID Control (Szafranski & Czyba, 2011), LQR Control (Priyambodo et al., 2020), Sliding Mode Control (Chen et al., 2016), Backstepping Control (Labbadi & Cherkaoui, 2019), Robust Control (Hasseni & Abdou, 2021), etc. However, most of the previously mentioned control methods suffer from limitations. Imprecise system modelling and non-modelled environmental disturbances may result in unacceptable performance or instability.

Today, artificial intelligence triggered a new wave of research in many fields (Jumper et al., 2021; Silver et al., 2017). The data-driven control methods can directly learn the corresponding control strategy from the interaction process of the controlled system so that it can adapt to new environments. Bansal et al. (2016) validates their proposed deep learning algorithm on a quadrotor testbed. On the other hand, reinforcement learning is a model-free algorithm widely used for control problems. Koch et al. (2019) present an intelligent high-precision flight control system using the reinforcement learning algorithm for UAVs. Moreover, the performance and accuracy of the internal control loop for quadrotor attitude control are analyzed and compared. Results indicate that the neural network has good generalization abilities and can learn the quadrotor dynamics accurately and apply them to the control system. Underwood & Husain (2010) propose an online parameter estimation, and the experimental results validate the effectiveness of the adaptive control method. O'Connell et al. (2022) have combined online adaptive learning with representation learning and adapted a DNN to learn a nonlinear representation. However, the environment's diversity is not considered in this work. Adapting to an environment completely different from the training set is challenging. Inspired by Shi et al. (2021), mechanical-based models with learnable dynamics and DNNs are constructed in this study for their interpretability and stability. We further investigate in this paper whether the robustness of the algorithm can be improved with OoD generalization methods.

### 2.2 OUT-OF-DISTRIBUTION GENERALIZATION

*Out-of-Distribution (OoD) generalization*, which involves generalizing under data distribution domain shifts, is an active research area in the community. Generalizing a prediction model under distribution shifts is the process of generalizing its performance. Many algorithms have been proposed to achieve the OoD generalization, including meta-learning (Li et al., 2019; Zhang et al., 2020), prototypical learning (Dubey et al., 2021), gradient alignment (Rame et al., 2022), domain adversarial learning (Akuzawa et al., 2019; Xu et al., 2020) and kernel methods (Li et al., 2018; Ghifary et al., 2016) etc.

Literature has extensively discussed how to deal with domain shift, and the OoD generalization problem is extensively studied in computer vision (Hsu et al., 2020), natural language processing (Hendrycks et al., 2020), speech recognition (Shankar et al., 2018), and other fields, but seldom in the context of online control. In Shi et al. (2021), a multi-task learning method for nonlinear systems was presented that can withstand disturbances and unknown environments.

Previous studies have suffered from shortcomings in lacking a discussion about the misspecification of dynamics systems and neglecting the gap between the simulation experiments and reality.

Additionally, the generalization of flight control is plagued by system measurement errors and unknown environmental parameters like wind modes and air density and resistance. In this work, we demonstrate the average control error can be upper-bounded by a constant when the environmental disturbances are within some range. Moreover, experimental results indicate that the *OoD-Control* is robust to shifts in the environmental domain.

## 3 PROBLEM FORMULATION

**Notations** We use subscripts (e.g., $t$ in $x_t^{(z)}$) to denote the time index and superscripts (e.g., $(z)$ in $x_t^{(z)}$) to denote the dynamics state under the environmental perturbation $z \sim \pi_0$, where $\pi_0$ is the environmental distribution, $\|\cdot\|$ denotes the 2-norm of a vector.

*Note:* Superscript $(z)$ in $x_t^{(z)}$ is used as a symbol to represent that the state is perturbed by the environmental distribution $z \sim \pi_0$ and $x_t^{(z)} = x_t + z$, where $x_t \in \mathbb{R}^n$ is the state in the windless environment.

In this paper, we consider a discrete nonlinear control system whose dynamics are described by the following formula:

$$x_{t+1}^{(z)} = f_0(x_t^{(z)}) + B(x_t^{(z)})u_t - f(x_t^{(z)}, c) + w_t, \quad 1 \le t \le T, \tag{1}$$

where $x_t^{(z)} \in \mathbb{R}^n$ is the state variable. $z \in \mathbb{R}^n$ changes with the environmental distribution domain shifts. $B(x_t^{(z)}) : \mathbb{R}^n \to \mathbb{R}^{n \times m}$ is the state-dependent actuation matrix; $u_t \in \mathbb{R}^m$ is the control input on the dynamic system; $f_0(x_t^{(z)}) : \mathbb{R}^n \to \mathbb{R}^n$ is the known nominal dynamic term that can be modelled with well-defined differential equations; $f(x_t^{(z)}, c) : \mathbb{R}^n \times \mathbb{R}^h \to \mathbb{R}^n$ are the unknown environment-dependent dynamics that are hard to be modelled and $c$ is the unknown environmental parameter, we also use $f(x_t^{(z)})$ for short in the following paragraphs; $w_t \in \mathbb{R}^n$ is the random noise vector.

We hypothesize that the environmental disturbance $w_t$ and the control $u_t$ are bounded. Due to the structural restrictions of the actuators, there are certain limits to the output. For example, the control output of the UAV is constrained by the maximum rotor blades' revolutions per minute (RPM). Here we give the *boundedness* Assumption.

**Assumption 1** *(**Bounded controls and disturbances**) Assume that the controller's output has an upper bound: $\forall t, \|u_t\| \le U$. Moreover, the environmental noise vectors are also bounded with zero expectations: $\forall t, \|w_t\| \le W, \mathbb{E}(w_t) = 0$.*

**Definition 1** *(**Average Control Error under disturbances**) The control error of the system under disturbance distribution $\pi_0$ at $t$ is calculated as $\|\mathbb{E}_{z \sim \pi_0}(x_t^{(z)}) - x_t^d\|$. The average control error $ACE_{\pi_0}$ of $T$ time steps is defined as the performance metric:*

$$ACE_{\pi_0} = \frac{1}{T} \sum_{t=1}^{T} \|\mathbb{E}_{z \sim \pi_0}(x_t^{(z)}) - x_t^d\| \tag{2}$$

*where $x_t^d$ denotes the desired states at $t$.*

**Remark 1** In Definition 1, we focus on fixed-point hovering and trajectory tracking. A sequence of perturbations matching $\pi_0$ is obtained by sampling for $N$ times: $(z_1, z_2, \cdots, z_N)$. And the perturbed states sequence is also derived at time $t$: $(x_t^{(z_1)}, x_t^{(z_2)}, \cdots, x_t^{(z_N)})$. $\|\mathbb{E}_{z \sim \pi_0}(x_t^{(z)})\|$ can be approximated with Monte Carlo method: $\sum_{i=1}^{N} x_t^{(z_i)}/N$, where the subscripts $i$ represent the index of the $i_{th}$ sample. Compared with the average control error definition in Shi et al. (2021); Åström & Murray (2008), $ACE_{\pi_0}$ represents the expectation of the difference between the actual states and the desired states of the dynamical system under environmental perturbations.

**Interaction protocol.** We set the study of the OoD adaptive flight control problem under the following interaction protocol:

1. Stochastically selects an environment for the controller to encounter every time step, which depends on the unobserved variable $c$ (e.g., wind condition and air density).

2. The controller interacts with the environment and observes the state $x_t^{(z)}$ to take action $u_t$.

3. Optionally changes $c$ after a short time and repeats from Step 1.

# 4 METHODOLOGY

We expect the UAV to learn the shared representation of the unknown dynamics between different environments so that it can generalize well in unseen areas with few adaptations. This section will introduce the methodology that provides the guaranteed upper bound for the prediction errors of the unknown dynamics.

**Notations and settings.** For modelling unknown environment-dependent dynamics, we use $\hat{f}(x_t^{(z)}, \hat{c}) = \mathbf{F}(\phi(x_t^{(z)}; \boldsymbol{\Theta}), \hat{c})$ inspired by (Shi et al., 2021). That means we consider the unknown environment-dependent dynamics typically consist of two coupling parts: the irregular higher-order aerodynamics of UAVs that are caused by the complex streamlined design, and the environment-dependent variables which encode wind field information. $\phi(x_t^{(z)}; \boldsymbol{\Theta})$ represents the former, which is a deep neural network (DNN) with $L$ layers parameterized by $\boldsymbol{\Theta}$. $\hat{c}$ is the latter, which is particular for a certain environment. This model enables us to consider the joint higher-order effects beyond nominal dynamics and enable agile control of UAVs. We use $\hat{f}(x_t^{(z)})$ for brevity in following paragraphs.

Assume an environmental distribution $\pi_0$, the perturbed unknown dynamics and the predictable value is denoted as the expectation of the states under environmental disturbances: $f_{\pi_0}(x_t^{(z)}) = \mathbb{E}_{z \sim \pi_0}[f(x_t^{(z)}, c)]$ and $\hat{f}_{\pi_0}(x_t^{(z)}) = \mathbb{E}_{z \sim \pi_0}[\mathbf{F}(\phi(x_t^{(z)}; \boldsymbol{\Theta}), \hat{c})] = \mathbb{E}_{z \sim \pi_0}[\hat{f}(x_t^{(z)}, \hat{c})]$. In OoD-control, the objective is to minimize the unknown dynamics prediction loss for controller inputs: $\ell_f = \|f_{\pi_0}(x_t^{(z)}) - \hat{f}_{\pi_0}(x_t^{(z)})\|$. The loss is mapped to $[0, 1]$ by $\tilde{h}(\|f_{\pi_0}(x_t^{(z)}) - \hat{f}_{\pi_0}(x_t^{(z)})\|)$ and we also use $\tilde{h}(x_t^{(z)})$ for short in the following paragraphs. $\tilde{h}(\cdot)$ satisfies: 1) $\tilde{h}(\cdot) \in [0, 1]$; 2) $\tilde{h}(\cdot)$ is monotonically decreasing and the inverse $\tilde{h}^{-1}(\cdot)$ exists. In the next section, we will propose a framework that provides a guaranteed upper bound for the ACE under OoD-Control.

## 4.1 THE PROOF OF ACE'S UPPER BOUND

Next, we will introduce a methodology to tackle the problem of generalization and give proof of ACE's upper bound. We want to verify that for any perturbation in $\mathcal{B} = \{\delta \in \mathbb{R}^n : \|\delta\|_2 \leq r\}$ with radius $r$, the lower bound of the prediction error maintains constant under unpredictable disturbance, i.e., $\forall \|\delta\| \leq r, \exists p > 0, \tilde{h}_{\pi_0}(x_t^{(z)}) > p$, it still holds $\tilde{h}_{\pi_0}(x_t^{(z)} + \delta) > p$. This conclusion is significant for the calculation of the upper bound for ACE. For a perturbation of radius r, the expectation of the prediction of the unknown dynamics remains the same.

Assume $\mathcal{H}$ is a function class, which includes $\tilde{h}_{\pi_0}(\cdot)$ and satisfies $\mathcal{H} = \{h : h(x) \in [0, 1], \forall x \in \mathbb{R}^n\}$. Performing the following optimization will result in a guaranteed lower bound. If $\mathcal{H}$ includes only $\tilde{h}_{\pi_0}(\cdot)$, the bound is exact:

$$\min_{\delta \in \mathcal{B}} \tilde{h}_{\pi_0}(x_t^{(z)} + \delta) \geq \min_{h \in \mathcal{H}} \min_{\delta \in \mathcal{B}} \left\{ h_{\pi_0}(x_t^{(z)} + \delta) \text{ s.t. } h_{\pi_0}(x_t^{(z)}) = \tilde{h}_{\pi_0}(x_t^{(z)}) \right\}. \qquad (3)$$

**Theorem 1 (Lagrangian)** $\mathcal{L}_{\pi_0}(\mathcal{H}, \mathcal{B})$ *is denoted as the lower bound in equation 3. Lagrangian methods can be adapted to solve inequality:*

$$\mathcal{L}_{\pi_0}(\mathcal{H}, \mathcal{B}) = \min_{h \in \mathcal{H}} \min_{\delta \in \mathcal{B}} \max_{\lambda \in \mathbb{R}} L(h, \delta, \lambda) \triangleq \min_{h \in \mathcal{H}} \min_{\delta \in \mathcal{B}} \max_{\lambda \in \mathbb{R}} \left\{ h_{\pi_0}(x_t^{(z)} + \delta) - \lambda[h_{\pi_0}(x_t^{(z)}) - \tilde{h}_{\pi_0}(x_t^{(z)})] \right\}.$$
$$(4)$$

*Exchanging the* min *and* max *yields the following dual form:*

$$\mathcal{L}_{\pi_0}(\mathcal{H}, \mathcal{B}) \geq \max_{\lambda \geq 0} \min_{h \in \mathcal{H}} \min_{\delta \in \mathcal{B}} L(h, \delta, \lambda) = \max_{\lambda \geq 0} \left\{ \lambda \tilde{h}_{\pi_0}(x_t^{(z)}) - \max_{\delta \in \mathcal{B}} \mathbb{D}_{\mathcal{H}}(\lambda_{\pi_0} \| \pi_\delta) \right\} \qquad (5)$$

*where* $\pi_\delta$ *represents the distribution of* $z + \delta$ *when* $z \sim \pi_0$ *and* $\mathbb{D}_{\mathcal{H}}(\lambda_{\pi_0} \| \pi_\delta) = \max_{h \in \mathcal{H}} \{\lambda \mathbb{E}_{z \sim \pi_0}[h(x_t^{(z)})] - \mathbb{E}_{z \sim \pi_\delta}[h(x_t^{(z)})]\} = \int [\lambda \pi_0(z) - \pi_\delta(z)]_+ dz.$

**Corollary 1** *(Gaussian noise)* *With Gaussian noise $\pi_0 = \mathcal{N}(0, \sigma^2 I)$ and bounded disturbance $\mathcal{B} = \{\delta : \|\delta\|_2 \leq r\}$, the lower bound in equation 5 satisfies:*

$$\mathcal{L}_{\pi_0}(\mathcal{H}, \mathcal{B}) = \max_{\lambda \geq 0} \left\{ \lambda \tilde{h}_{\pi_0}(x_t^{(z)}) - \max_{\delta \in \mathcal{B}} \mathbb{D}_{\mathcal{H}}(\lambda_{\pi_0} || \pi_\delta) \right\} \geq \Phi(\Phi^{-1}(\tilde{h}_{\pi_0}(x_t^{(z)}) - \frac{r}{\sigma}) \quad (6)$$

*where $\Phi(\cdot)$ represents the Gaussian Cumulative Density Function (CDF). For the case p=0.5, i.e., $\tilde{h}_{\pi_0}(x_t^{(z)}) > 0.5$, the radius satisfies $r \leq \sigma \Phi^{-1}(\tilde{h}_{\pi_0}(x_t^{(z)}))$. As a side note, the Monte Carlo method for perturbation radius calculation is also given by Algorithm 2 in Appendix B.3.*

## 4.2 Average Control Error Bound

**The selection of $\tilde{h}$.** Given a sequence of state variables at $t$ under perturbation $z \sim \pi_0 = \mathcal{N}(0, \sigma^2)$: $X = (x_t^{(z_1)}, x_t^{(z_2)}, \ldots, x_t^{(z_N)})$. Moreover, the predicted and *unknown dynamics* sequences are defined as $F_p = (\hat{f}(x_t^{(z_1)}), \hat{f}(x_t^{(z_2)}), \ldots, \hat{f}(x_t^{(z_N)}))$ and $F_u = (f(x_t^{(z_1)}), f(x_t^{(z_2)}), \ldots, f(x_t^{(z_N)}))$. Let $D_p$ be the discrepancy sequence between $F_p$ and $F_u$. $D_p = (\|\hat{f}(x_t^{(z_1)}) - f(x_t^{(z_1)})\|, \|\hat{f}(x_t^{(z_2)}) - f(x_t^{(z_2)})\|, \ldots, \|\hat{f}(x_t^{(z_N)}) - f(x_t^{(z_N)})\|)$. Denote by $\hat{p}$ the successful rate for the prediction error under a given threshold $\varepsilon_t$. By simulating with a large sample size $N$, $\hat{p}$ is calculated as:

$$\hat{p} \triangleq \mathbb{P}(\|\hat{f}(x_t^{(z)}) - f(x_t^{(z)})\| < \varepsilon_t) = \frac{n_a}{N} \quad (7)$$

where $n_a$ is the number of elements in $D_p$ less than $\varepsilon_t$. Recalling the $\tilde{h}$ function's requirements, we can instantiate $\tilde{h}$ as follows:

$$\tilde{h}(\|\hat{f}(x_t^{(z)}) - f(x_t^{(z)})\|) \triangleq \hat{p} - k\sqrt{\frac{\hat{p}(1 - \hat{p})}{n}} \quad (8)$$

where $k = \Phi^{-1}(1 - \frac{\alpha}{2})$ is the $1 - \frac{\alpha}{2}$ quantile of a standard normal distribution. Moreover, this equation represents the lower confidence bound estimation of the error under a given confidence level $\alpha$. As noise increases, the predicted value will deviate from the actual value by a more significant amount. Therefore, the $\tilde{h}$ will decrease monotonically. Besides, the range of the lower confidence bound lies in $[0, 1]$, which satisfies both requirements for $\tilde{h}$ as discussed in the previous section.

Let $\underline{b}$ denote the lower confidence bound in equation 8. In section 4.1, a proof is given that $\tilde{h}_{\pi_0}(x_t^{(z)})$ and $\tilde{h}_{\pi_0}(x_t^{(z)} + \delta)$ has the same lower bound under the disturbances $\|\delta\| \leq r$. The radius ensuring an equal lower bound under the perturbation in this paper is:

$$r = \sigma \Phi^{-1}(\underline{b}) = \sigma \Phi^{-1}(\hat{p} - k\sqrt{\frac{\hat{p}(1 - \hat{p})}{n}}). \quad (9)$$

**Controlling.** The control term $u$ consists of three parts: feedback, feedforward, and residual, where the feedback part gets information from sensors and minimizes the gap between $x_t^{(z)}$ and $x_t^d$, the feedforward part offsets the nominal term $f_0(x_t^{(z)})$, and the residual part counterweights unknown environment-dependent term $f(x_t^{(z)})$. $B^\dagger(x_t^{(z)})$ is the pseudo-inverse $B(x_t^{(z)})$.

The controller of model-based control is

$$u_t = B^\dagger(x_t^{(z)})(-f_0(x_t^{(z)}) + \hat{f}(x_t^{(z)})). \quad (10)$$

Thus, the equation 1 becomes:

$$x_{t+1} = \hat{f}(x_t^{(z)}) - f(x_t^{(z)}) + w_t. \quad (11)$$

**Lemma 1** *(ACE bound in ideal case)* *For any perturbation in $\mathcal{B} = \{\delta : \|\delta\|_2 \leq r\}$, the theoretical average control error is bounded as:*

$$ACE_{\pi_\delta} = \frac{1}{T} \sum_{t=1}^{T} \|\mathbb{E}_{z \sim \pi_\delta}[\hat{f}(x_t^{(z)})] - \mathbb{E}_{z \sim \pi_\delta}[f(x_t^{(z)})]\| \leq \tilde{h}^{-1}(p). \quad (12)$$

**Remark 2** Average control error is related to the prediction error of unknown dynamics and environmental perturbations in the ideal case. Compared with the upper bound calculated in Shi et al. (2021), the derived bound is more general. Our calculation method does not require the system to be fully actuated and does not require the nominal dynamics to be exponentially input-to-state stable (e-ISS). The assumption of e-ISS is too strong. And many dynamic systems are under-actuated in the real world, such as quadrotors.

**Corollary 2** *(ACE bound under control actuation misspecification)* $\Delta B(x_t^{(z)})$ *is the parametric misspecification in the actuation matrix, and the* $ACE_{\pi_\delta}$ *satisfies:*

$$ACE_{\pi_\delta} = \frac{1}{T} \sum_{t=1}^{T} \|\mathbb{E}_{z \sim \pi_\delta}(x_t^{(z)}) - x_t^d\| \le \tilde{h}^{-1}(p) + \frac{1}{T} \sum_{t=1}^{T} \|\mathbb{E}_{z \sim \pi_\delta}[\Delta B(x_t^{(z)}) B^\dagger(x_t^{(z)}) e_f(x_t^{(z)})]\|$$

*where* $e_f(x_t^{(z)}) = \hat{f}(x_t^{(z)}, c) - f_0(x_t^{(z)})$.

**Lemma 2** *(Trajectory tracking ACE of quadrotor)* *In this paper, for environmental disturbances in* $\mathcal{B} = \{\delta : \|\delta\|_2 \le r\}$, *the quadrotors' trajectory tracking error formula is given as:*

$$ACE_{\pi_\delta} = \frac{1}{T} \sum_{t=1}^{T} \|\mathbb{E}_{z \sim \pi_\delta}(x_t^{(z)}) - x_t^d\| = \frac{1}{T} \sum_{t=1}^{T} \|\mathbb{E}_{z \sim \pi_\delta}[C_1 e^{r_1 x_t^{(z)}} + C_2 e^{r_2 x_t^{(z)}}] - \tilde{h}^{-1}(p)/K_v\| \tag{13}$$

*where* $C_1 = -x_t^d - C_2 + \epsilon/K_v$, $C_2 = (\epsilon + (K_v x_t^d - \epsilon)e^{r_1 x_t^d})/K_v(e^{r_2 x_t^d} - e^{r_1 x_t^d})$ *and* $\epsilon = f(x_t^{(z)}, c) - \hat{f}(x_t^{(z)}, \hat{c})$, *which is the prediction error of the unknown dynamics.*

**Remark 3** Note that Lemma 2 gives the trajectory tracking error that can be calculated. Based on the previous description, it was demonstrated that the errors for unknown dynamics under perturbation have the same bound. The detailed proof can be found in Appendix A.5.

### 4.3 OOD GENERALIZATION ALGORITHM

Based on the theoretical analysis above, we propose an algorithm—out-of-distribution generalization for adaptive flight control named *OoD-Control*. We focus on minimizing the prediction loss and learning $\Theta$ during the simulation. We intend to design an OoD-controller with lower ACEs that converges the estimated *unknown dynamics* $\hat{f}(x_t^{(z)})$ faster to the true dynamics $f(x_t^{(z)})$ under environment distribution domain shifts.

The proposed OoD-Control algorithm is shown in **Algorithm** 1 (see Appendix B.1). Given a set of distribution functions $X$, $\chi \in X$ are picked for each iteration. The wind velocity is a series of random variables sampled from $\chi$. (Specifically, we use $X$ for the training distribution set with each member denoted as $\chi$. For the testing set, we use $\Omega$ and $\omega$ instead.) Each time-series simulation begins with random noise $\epsilon_1$ being introduced to the structural parameters of the system. At each iteration, the predicted loss, which measures the error between the unknown dynamics and its prediction is minimized. After the unknown dynamic predictor is trained, it can be used for model-based control as discussed in Section 4.2.

## 5 EXPERIMENTS AND RESULTS

In this section, numerical experiments on the inverted pendulum and quadrotors will be conducted to demonstrate the effectiveness of the proposed OoD-control algorithmic framework. To better understand the proposed OoD-Control algorithm and environment setting, we choose an uncoupled dynamics model, the inverted pendulum, as the introductory example before the quadrotor instance.

### 5.1 DYNAMICS MODELING

**Inverted Pendulum.** Consider an inverted pendulum, the dynamic model of the pendulum is:

$$ml^2\ddot{\theta} - mlg\sin\theta = u + f(\theta, \dot{\theta}, c) \tag{14}$$

where $\theta$ represents the angle away from the center, $l$ is the length of the arm, $g$ is gravitational acceleration, $l$ is the length of the pendulum's arm, and $m$ is the mass. The state variable consists of $\theta$ and $\dot{\theta}$, which could be measured by position and inertia sensors. $f(\cdot)$ represents the unknown dynamic term, including air resistance, wind force and modelling misspecification, subject to $\theta$, $\dot{\theta}$ and environment parameter $c$. $u$ is the controlling term. Our goal is to keep the pendulum closer to the center, i.e., minimize average control error.

**Quadrotor.** The quadrotor is a plane model where the four rotors are always on the same plane. So the quadrotor adjusts its attitude by setting different rotation speeds for the four rotors. We define the dynamic model of quadrotor as:

$$m\dot{v} = mg + R(\theta)f_T + f \tag{15}$$

$$J\ddot{\theta} = J\dot{\theta} \times \dot{\theta} + \tau \tag{16}$$

where $\theta$ represents the attitude angel of the quadrotor; $R(\theta) \in \mathbb{R}^{3\times3}$ is the attitude rotation matrix subject to $\theta$; $J$ is the inertia matrix of the quadrotor, $f_T$ is the force imposed on the system; $\tau$ is the total torque; $m$ is the mass of quadrotor and $g$ is the gravitational acceleration; $f_T$ and $\tau$ are subject to the speeds of rotors $n_r \in \mathbb{R}^{1\times4}$. In the experiments, the goal is to maintain the quadrotor's position states or follow a given trajectory under turbulent environments.

## 5.2 COMPARISON METHOD

In the experiments, the proposed adaptive UAV flight control algorithm *OoD-Control* are compared with *OMAC* (Shi et al., 2021) and *no-adapt (PID)* method. *OMAC* (online meta-learning adaptive control) is the state-of-the-art data-driven UAV flight control method. In the *OMAC* paper, three versions of OMAC are provided with different model specifications: convex, bi-convex, and deep learning. We illustrate the results of deep learning because it is the best-performing version of OMAC.

Meanwhile, the *no-adapt* method and the *omniscient* method are compared in this paper. *No-adapt* indicates the controller cannot perceive the environmental domain shifts with $\hat{f}(x_t^{(z)}) = 0$ which is just the conventional *PID* controller. *omniscient* is the controller which has access to the unknown dynamics perfectly, i.e., $\hat{f}(x_t^{(z)}) = f(x_t^{(z)})$. Among all the controllers, *no-adapt* and *omniscient* are the two extremes, with no-adapt being unable to predict while omniscient can do so with zero error. We run each simulation ten times with different random seeds to obtain the mean and standard deviation of ACE under perturbation for rigorousness.

## 5.3 WIND FIELD CONSTRUCTION AND FLIGHT TRAJECTORY DESIGN

Wind fields can be derived according to the *Navier-Stokes (N-S) equations* and the *continuum equation*. However, in practice, the N-S equations are generally hard to be solved due to their high computational cost. For turbulent wind field simulations, the Dryden model (Specification, 1980) is widely used. We refer to the Dryden model to simulate turbulent wind fields on quadrotors by generating Gaussian wind disturbances (Beal, 1993).

To construct realistic situations in the inverted pendulum and quadrotor experiment, we simulated two types of winds: *turbulent wind* and *gust*. For turbulent winds, the speed and direction change at any time. In the case of gusts, the wind speed remains constant over a period of time. For further study, we divided the two wind fields *turbulent wind* and *gust* into three categories respectively according to their strength: *breeze*, *strong breeze* and *gale*. The direction and strength of the turbulent winds change continuously and the wind forces are applied to the object. This requires higher manoeuvrability to maintain stability. The wind environment setting in the experiment can be found in Appendix B.6.1.

Quadrotors must also be capable of flying along the desired trajectory and hovering at a fixed point for various applications, such as inspection, patrol, and delivery. In order to meet the requirements of different application scenarios, we design a variety of trajectories to test the performance of the proposed OoD-Control under different situations. The designed trajectory can correspond to a specific application scenario, *hovering* for fixed-point photography, *figure-8* trajectories for scenarios

requiring high manoeuvrability, the *spiral trajectory* for power lines detection, and *sin-forward* for transporting items in the forests or area scanning. The mathematical forms of the trajectories are shown in Appendix B.6.1.

## 5.4 RESULTS

**Pendulum.** Table 1 (see Appendix B.2) and Figure 1 illustrate the average and standard deviation of the control errors in different testing environments. We mainly compare the OoD-Control algorithm with the OMAC. And for the completeness of the experiments, we also set two control groups: the no-adapt and the omniscient.

As shown in row 3 of Table 1, our OoD-Control algorithm performs significantly better than the OMAC in the gale dataset. That means the former generalizes better than the latter when meeting a large environmental distribution shift. When changing to the less difficult dataset such as the strong breeze, we can see that the gap between the two algorithms decreases, but OoD-Control still achieves nearly half the score of the OMAC. (The breeze dataset is not complicated enough to distinguish the mentioned methods.

We also show the results of when $\hat{c}$ is unchanged. In this setting, both OMAC and OoD-Control perform terribly because the variable used to fit the ground truth of $c$ is frozen.

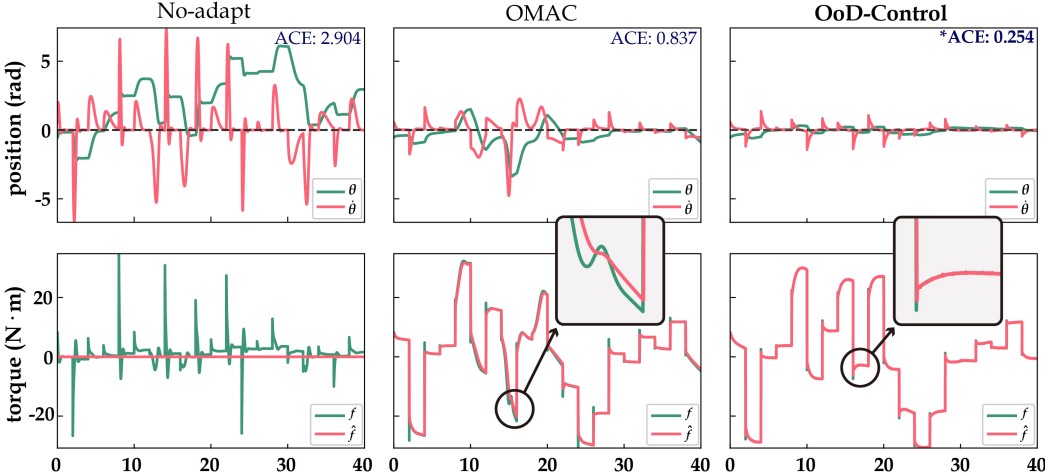

Figure 1: Result of inverted pendulum experiment where the testing environment is Gale. The black dashed line represents the desired states for the inverted pendulum. The objective of this task is to maintain the angle $\theta$ and angular velocity $\dot{\theta}$ of the inverted pendulum to zero. $f$ is the ground truth of the torque, while $\hat{f}$ is the predicted torque. $*$ is given for the best performance. As shown in the amplified areas (the black rounded rectangles), our algorithm predicts much better than OMAC.

**Quadrotor.** We show the result of the quadrotor task in Figure 2 and Table 2 (see Appendix B.6.1). As testing environments differ from training ones, the OoD-Control method maintains good stability and tracking accuracy. In some cases, the performance of OoD-Control goes close to the omniscient case where the rotor is provided with precise wind conditions, which shows that our algorithm is able to predict the wind with acceptable error. Our algorithm achieves lower ACE than baseline methods (60% than OMAC and over 70% than PID) in most difficult cases. Besides, we shorten the training time to test its sample efficiency, and it turns out that our algorithm performs well in few-shot learning. By adding noise during training DNN and fixing a learning rate of meta-learning of $\hat{c}$, our algorithm gains robustness and adapts quicker in a different environment.

We tested our method under several different trajectories, and OoD-Control outperforms the baseline and conventional no-adapt methods when the distribution domain shifts during the testing process. Meanwhile, OoD-control can learn more from changes in the environment and apply it as prior knowledge, thereby improving its adaptability.

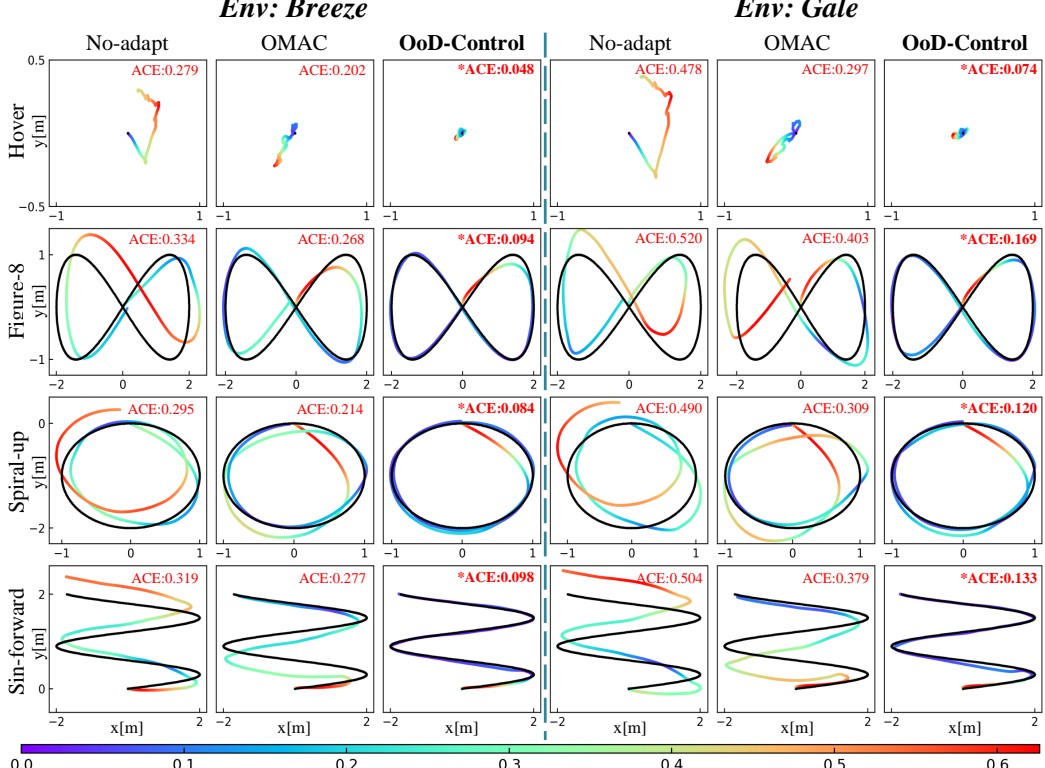

Figure 2: 2D view of trajectories in different wind conditions and performance comparison of OMAC and OoD-Control algorithms for trajectory tracking. The goal of the controller is to get closer to the desired trajectory (black line). Different colors demonstrate the distance from the actual location to the desired location, corresponding to the color bar at the bottom. The shade of color indicates the magnitude of the deviation in position. As compared to OMAC and no-adapt, the proposed OoD-Control method provides more accurate results across a wide range of wind environments and trajectories. The average ACE of ten independent experiments is marked in the subplot and ∗ is given for the best performance under the same environment.

The OMAC and OoD-Control algorithm were tested under hovering, figure-8, spiral upward and sin-forward trajectory scenarios. Figure 2 and Table 2 (see Appendix B.6.1) show the trajectory tracking experimental results. OoD-Control provides more accurate results across trajectories under a wide range of wind environments and achieves state-of-the-art performance in all these situations compared with the baseline.

Based on experiments, it has been demonstrated that systems trained by the proposed OoD-Control algorithm perform state-of-the-art. In addition, the control method can be applied to different dynamical models and is extensible and universally applicable.

## 6    CONCLUSION

In this paper, we theoretically demonstrate that the average control error is upper-bounded by a constant when the perturbation on the state variables is within a certain radius for UAV flight control. Besides, we propose an algorithmic framework—OoD-Control that is evaluated under turbulent environmental conditions. Based on the results of our experiments, we can conclude that our algorithm is scalable and pervasively applicable that can be applied to a variety of dynamic models. For future work, we will explore extending our algorithmic framework to more UAV types, such as unmanned helicopters, tilt-rotors, and unmanned fixed-wing aircraft. As far as we are aware, this is one of the first papers that theoretically discusses out-of-distribution problems in the context of online adaptive UAV flight control.

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

# A  PROOF OF LEMMA AND COROLLARY

## A.1  PROOF FOR THEOREM 1

**Theorem 1**   $\mathcal{L}_{\pi_0}(\mathcal{H}, \mathcal{B})$ *is denoted as the lower bound in equation 3. Lagrangian methods can be adapted to solve inequality.*

$$\mathcal{L}_{\pi_0}(\mathcal{H}, \mathcal{B}) = \min_{h \in \mathcal{H}} \min_{\delta \in \mathcal{B}} \max_{\lambda \in \mathbb{R}} L(h, \delta, \lambda) \triangleq \min_{h \in \mathcal{H}} \min_{\delta \in \mathcal{B}} \max_{\lambda \in \mathbb{R}} \left\{ h_{\pi_0}(x_t^{(z)} + \delta) - \lambda[h_{\pi_0}(x_t^{(z)}) - \tilde{h}_{\pi_0}(x_t^{(z)})] \right\}$$
(17)

*Exchanging the* min *and* max *yields the following dual form:*

$$\mathcal{L}_{\pi_0}(\mathcal{H}, \mathcal{B}) \geq \max_{\lambda \geq 0} \min_{h \in \mathcal{H}} \min_{\delta \in \mathcal{B}} L(h, \delta, \lambda) = \max_{\lambda \geq 0} \left\{ \lambda \tilde{h}_{\pi_0}(x_t^{(z)}) - \max_{\delta \in \mathcal{B}} \mathbb{D}_{\mathcal{H}}(\lambda_{\pi_0} || \pi_\delta) \right\}$$
(18)

*where  $\pi_\delta$  represents the distribution of  $z + \delta$  when  $z \sim \pi_0$  and  $\mathbb{D}_{\mathcal{H}}(\lambda_{\pi_0} || \pi_\delta) = \max_{h \in \mathcal{H}} \{ \lambda \mathbb{E}_{z \sim \pi_0}[h(x_t^{(z)})] - \mathbb{E}_{z \sim \pi_\delta}[h(x_t^{(z)})] \} = \int [\lambda \pi_0(z) - \pi_\delta(z)]_+ dz.$*

*Proof.*

(i)

$$
\begin{aligned}
\mathcal{L}_{\pi_0}(\mathcal{H}, \mathcal{B}) &= \min_{h \in \mathcal{H}} \min_{\delta \in \mathcal{B}} \max_{\lambda \in \mathbb{R}} \left\{ h_{\pi_0}(x_t^{(z)} + \delta) - \lambda[h_{\pi_0}(x_t^{(z)}) - \tilde{h}_{\pi_0}(x_t^{(z)})] \right\} \\
&\geq \max_{\lambda \geq 0} \min_{h \in \mathcal{H}} \min_{\delta \in \mathcal{B}} \left\{ h_{\pi_0}(x_t^{(z)} + \delta) - \lambda[h_{\pi_0}(x_t^{(z)}) - \tilde{h}_{\pi_0}(x_t^{(z)})] \right\} \\
&= \max_{\lambda \geq 0} \left\{ \lambda \tilde{h}_{\pi_0}(x_t^{(z)}) - \max_{h \in \mathcal{H}} (\lambda h_{\pi_0}(x_t^{(z)}) - h_{\pi_\delta}(x_t^{(z)})) \right\} \\
&= \max_{\lambda \geq 0} \left\{ \lambda \tilde{h}_{\pi_0}(x_t^{(z)}) - \max_{\delta \in \mathcal{B}} \mathbb{D}_{\mathcal{H}}(\lambda_{\pi_0} || \pi_\delta) \right\}
\end{aligned}
$$

(ii) We denote the sign function  $sgn(z)$  as:

$$sgn(\delta) = \begin{cases} 1, \text{if } [\lambda \pi_0(z) - \pi_\delta(z)] \geq 0 \\ 0, \text{if } [\lambda \pi_0(z) - \pi_\delta(z)] < 0 \end{cases}$$
(19)

Thus we can calculate  $\mathbb{D}_{\mathcal{H}}(\lambda_{\pi_0} || \pi_\delta)$  directly as:

$$
\begin{aligned}
\mathbb{D}_{\mathcal{H}}(\lambda_{\pi_0} || \pi_\delta) &= \max_{h \in \mathcal{H}} (\lambda h_{\pi_0}(x_t^{(z)}) - h_{\pi_0}(x_t^{(z)} + \delta)) \\
&= \max_{h \in \mathcal{H}} \left\{ \lambda \mathbb{E}_{z \sim \pi_0}[h(x_t^{(z)})] - \mathbb{E}_{z \sim \pi_\delta}[h(x_t^{(z)})] \right\} \\
&= \int sgn(x_t^{(z)})[\lambda \pi_0(z) - \pi_\delta(z)] dz \\
&= \int [\lambda \pi_0(z) - \pi_\delta(z)]_+ dz
\end{aligned}
$$

## A.2  PROOF FOR COROLLARY 1

**Corollary 1**   *(Gaussian noise) With Gaussian noise  $\pi_0 = \mathcal{N}(0, \sigma^2 I)$  and bounded disturbance  $\mathcal{B} = \{\delta : ]\|\delta\|_2 \leq r\}$ , the lower bound in equation 5 satisfies:*

$$\mathcal{L}_{\pi_0}(\mathcal{H}, \mathcal{B}) = \max_{\lambda \geq 0} \left\{ \lambda \tilde{h}_{\pi_0}(x_t^{(z)}) - \max_{\delta \in \mathcal{B}} \mathbb{D}_{\mathcal{H}}(\lambda_{\pi_0} || \pi_\delta) \right\} \geq \Phi(\Phi^{-1}(\tilde{h}_{\pi_0}(x_t^{(z)})) - \frac{r}{\sigma})$$
(20)

*where  $\Phi(\cdot)$  represents the Gaussian Cumulative Density Function (CDF). For the case p=0.5, i.e.,  $\tilde{h}_{\pi_0}(x_t^{(z)}) > 0.5$ , the radius satisfies  $r \leq \sigma \Phi^{-1}(\tilde{h}_{\pi_0}(x_t^{(z)}))$ .*

*Proof.*

$$\mathcal{L} \geq \Phi(\Phi^{-1}(h_{\pi_0}(x_t^{(z)})) - \frac{r}{\sigma}) > \frac{1}{2} \tag{21}$$

$$\mathcal{L} \geq \min_{\|\delta\| \leq r} \max_{\lambda \geq 0} \left\{ \lambda \tilde{h}_{\pi_0}(x_t^{(z)}) - \int [\lambda \pi_0(z) - \pi_\delta(z)]_+ dz \right\} \tag{22}$$

We denote $C_\lambda = \{z : \lambda \pi_0(z) \geq \pi_\delta(z)\} = \{z : \delta^T z \leq \frac{\|\delta\|^2}{2} + \sigma^2 \ln \lambda\}$ and $F(\delta, \lambda) = \lambda \tilde{h}_{\pi_0}(x_t^{(z)}) - \int [\lambda \pi_0(z) - \pi_{\delta a}(z)]_+ dz$. Then we will get:

$$
\begin{aligned}
F(\delta, \lambda) &= \lambda \tilde{h}_{\pi_0}(x_t^{(z)}) - \int [\lambda \pi_0(z) - \pi_\delta(z)]_+ dz \\
&= \lambda \tilde{h}_{\pi_0}(x_t^{(z)}) - \int_{C_\lambda} [\lambda \pi_0(z) - \pi_\delta(z)] dz \\
&= \lambda \tilde{h}_{\pi_0}(x_t^{(z)}) - \lambda \Phi\left(\frac{\|\delta\|_2}{2\sigma} + \frac{\sigma \ln \lambda}{\|\delta\|_2}\right) + \Phi\left(\frac{-\|\delta\|_2}{2\sigma} + \frac{\sigma \ln \lambda}{\|\delta\|_2}\right)
\end{aligned}
$$

It is notable that $F(\delta, \lambda)$ is a concave function w.r.t $\lambda$, thus the maximum value occurs when $\frac{\partial F(\delta, \lambda)}{\partial \lambda}\big|_{\lambda = \lambda_\delta} = 0$. A direct calculation gives $\lambda_\delta = \exp\left(\frac{2\sigma \|\delta\|_2 \Phi^{-1}(h_{\pi_0}(x_t^{(z)})) - \|\delta\|_2^2}{2\sigma^2}\right)$. There is

$$
\begin{aligned}
\mathcal{L} &\geq \min_{\|\delta\| \leq r} \max_{\lambda \geq 0} F(\delta, \lambda) \\
&= \min_{\|\delta\| \leq r} \Phi\left(\frac{-\|\delta\|_2}{2\sigma} + \frac{\sigma \ln \lambda_\delta}{\|\delta\|_2}\right) \\
&= \min_{\|\delta\| \leq r} \Phi\left(\Phi^{-1}(h_{\pi_0}(x_t^{(z)})) - \frac{\|\delta\|_2}{\sigma}\right) \\
&= \Phi\left(\Phi^{-1}(h_{\pi_0}(x_t^{(z)})) - \frac{r}{\sigma}\right)
\end{aligned}
$$

In case $p=0.5$, the perturbation radius $r$ is calculated as [1]:

$$\min_{\|\delta\| \leq r} \max_{\lambda \geq 0} F(\delta, \lambda) > \frac{1}{2} \Leftrightarrow \Phi\left(\Phi^{-1}(h_{\pi_0}(x_t^{(z)})) - \frac{r}{\sigma}\right) > \frac{1}{2} \Leftrightarrow r < \sigma \Phi^{-1}(h_{\pi_0}(x_t^{(z)})) \tag{23}$$

### A.3    PROOF FOR LEMMA 1

**Lemma 1**    *(ACE bound in ideal case) For any perturbation in $\mathcal{B} = \{\delta : \|\delta\|_2 \leq r\}$, the theoretical average control error is bounded as:*

$$ACE_{\pi_\delta} = \frac{1}{T} \sum_{t=1}^{T} \|\mathbb{E}_{z \sim \pi_\delta}[\hat{f}(x_t^{(z)})] - \mathbb{E}_{z \sim \pi_\delta}[f(x_t^{(z)})]\| \leq \tilde{h}^{-1}(p) \tag{24}$$

*Proof.*
From equation 1, we consider a discrete nonlinear control-affine system:

$$x_{t+1} = f_0(x_t) + B(x_t)u_t - f(x_t, c) + w_t, \quad 1 \leq t \leq T, \tag{25}$$

The controller of model-based control with the ideal model is

$$u_t = B^\dagger(x_t^{(z)})(-f_0(x_t^{(z)}) + \hat{f}(x_t^{(z)})) \tag{26}$$

thus, the equation 1 becomes:

$$x_{t+1} = \hat{f}(x_t^{(z)}) - f(x_t^{(z)}, c) + w_t \tag{27}$$

---

[1]The calculation method of the radius $r$ is given in Cohen et al. (2019) for the case p=0.5.

Following a straightforward calculation, we will obtain the upper ACE bound under disturbance $z \sim \pi_\delta$

$$ACE_{\pi_\delta} = \frac{1}{T}\sum_{t=1}^{T}\|\mathbb{E}_{z\sim\pi_\delta}(x_t^{(z)}) - x_t^d\| = \frac{1}{T}\sum_{t=1}^{T}\|\mathbb{E}_{z\sim\pi_\delta}[\hat{f}(x_t^{(z)})] - \mathbb{E}_{z\sim\pi_\delta}[f(x_t^{(z)}, c)] + \mathbb{E}_{z\sim\pi_\delta}(w_t)\|$$

$$\leq \frac{1}{T}\sum_{t=1}^{T}\|\mathbb{E}_{z\sim\pi_\delta}[\hat{f}(x_t^{(z)})] - \mathbb{E}_{z\sim\pi_\delta}[f(x_t^{(z)}, c)]\| + \|\mathbb{E}_{z\sim\pi_\delta}(w_t)\|$$

$$\leq \|\hat{h}^{-1}(p)\| \tag{28}$$

## A.4 Proof for Corollary 2

**Corollary 2** *(ACE bound under control actuation misspecification) $\Delta B(x_t^{(z)})$ is the parametric misspecification in the actuation matrix, and the $ACE_{\pi_\delta}$ satisfies:*

$$ACE_{\pi_\delta} = \frac{1}{T}\sum_{t=1}^{T}\|\mathbb{E}_{z\sim\pi_\delta}(x_t^{(z)}) - x_t^d\| \leq \tilde{h}^{-1}(p) + \frac{1}{T}\sum_{t=1}^{T}(\|\mathbb{E}_{z\sim\pi_\delta}[\Delta B(x_t^{(z)})B^\dagger(x_t^{(z)})e_f(x_t^{(z)})]\|)$$

*where $e_f(x_t^{(z)}) = \hat{f}(x_t^{(z)}, c) - f_0(x_t^{(z)})$.*

*Proof.*

$$x_{t+1} = f_0(x_t^{(z)}) + (B(x_t^{(z)}) + \Delta B(x_t^{(z)}))u(t) - f(x_t^{(z)}, c) + w_t$$
$$= \hat{f}(x_t^{(z)}) - f(x_t^{(z)}, c) + w_t + \Delta B(x_t^{(z)})B^\dagger(x_t^{(z)})[-f_0(x_t^{(z)}) + \hat{f}(x_t^{(z)})]$$
$$= \hat{f}(x_t^{(z)}) - f(x_t^{(z)}, c) + w_t + \Delta B(x_t^{(z)})B^\dagger(x_t^{(z)})e_f(x_t^{(z)}) \tag{29}$$

We use $e_f(x_t^{(z)})$ to denote $\hat{f}(x_t^{(z)}) - f_0(x_t^{(z)})$, the ACE upper bound is calculated as follows:

$$ACE_{\pi_\delta} = \frac{1}{T}\sum_{t=1}^{T}\|\mathbb{E}_{z\sim\pi_\delta}(x_t^{(z)}) - x_t^d\|$$

$$= \frac{1}{T}\sum_{t=1}^{T}\|\mathbb{E}_{z\sim\pi_\delta}[\hat{f}(x_t^{(z)}) - f(x_t^{(z)}, c) + w_t + \Delta B(x_t^{(z)})B^\dagger(x_t^{(z)})e_f(x_t^{(z)})]\|$$

$$= \frac{1}{T}\sum_{t=1}^{T}\|\mathbb{E}_{z\sim\pi_\delta}[\hat{f}(x_t^{(z)}) - f(x_t^{(z)}, c)] + \mathbb{E}_{z\sim\pi_0}(w_t) + \mathbb{E}_{z\sim\pi_\delta}[\Delta B(x_t^{(z)})B^\dagger(x_t^{(z)})e_f(x_t^{(z)})]\|$$

$$= \frac{1}{T}\sum_{t=1}^{T}\|\mathbb{E}_{z\sim\pi_\delta}[\hat{f}(x_t^{(z)})] - \mathbb{E}_{z\sim\pi_\delta}[f(x_t^{(z)}, c)] + \mathbb{E}_{z\sim\pi_\delta}[\Delta B(x_t^{(z)})B^\dagger(x_t^{(z)})e_f(x_t^{(z)})]\|$$

$$\leq \tilde{h}^{-1}(p) + \frac{1}{T}\sum_{t=1}^{T}(\|\mathbb{E}_{z\sim\pi_\delta}[\Delta B(x_t^{(z)})B^\dagger(x_t^{(z)})e_f(x_t^{(z)})]\|) \tag{30}$$

## A.5 Proof for Lemma 2

**Lemma 2** *(Trajectory tracking ACE of quadrotor) In this paper, for environmental disturbances in $\mathcal{B} = \{\delta : \|\delta\|_2 \leq r\}$, the formula of quadrotors' trajectory tracking error is given as:*

$$ACE_{\pi_\delta} = \frac{1}{T}\sum_{t=1}^{T}\|\mathbb{E}_{z\sim\pi_\delta}(x_t^{(z)}) - x_t^d\| = \frac{1}{T}\sum_{t=1}^{T}\|\mathbb{E}_{z\sim\pi_\delta}[C_1 e^{r_1 x_t^{(z)}} + C_2 e^{r_2 x_t^{(z)}} - \epsilon/K_v]\|$$

$$= \frac{1}{T}\sum_{t=1}^{T}\|\mathbb{E}_{z\sim\pi_\delta}[C_1 e^{r_1 x_t^{(z)}} + C_2 e^{r_2 x_t^{(z)}}] - \tilde{h}^{-1}(p)/K_v\|$$

$$\tag{31}$$

where $C_1 = -x_t^d - C_2 + \epsilon/K_v$, $C_2 = (\epsilon + (K_v x_t^d - \epsilon)e^{r_1 x_t^d})/K_v(e^{r_2 x_t^d} - e^{r_1 x_t^d})$ and $\epsilon = f(x_t^{(z)}, c) - \hat{f}(x_t^{(z)}, \hat{c})$, which is the prediction error of the unknown dynamics.

*Proof.*

Recall the kinetic function in equation 15: $m\dot{v} = mg + Rf_T + f_t$. The desired control force $f_d$ is designed as:

$$\begin{cases} f_d = Rf_T = \bar{f}_d - \hat{f}_t \\ \bar{f}_d = m\dot{v}_r + K_v x_e - mg \end{cases} \tag{32}$$

where $x_e = x_t^{(z)} - x_t^d$. $x_e$ is the error of trajectory tracking and $v_r$ is the desired velocity at $t$. By substituting equation 32 into equation 15, the UAV dynamics becomes:

$$m\dot{v} = mg + Rf_T + f_t$$
$$m\dot{v} - mg - \bar{f}_d = f_t - \hat{f}_t$$
$$m(\dot{v} - \dot{v}_r) - K_v x_e = f_t - \hat{f}_t$$
$$m\ddot{x}_e - K_v x_e - \epsilon = 0$$
$$\ddot{x}_e - \frac{K_v}{m}x_e - \frac{1}{m}\epsilon = 0 \tag{33}$$

where $\epsilon = f_t - \hat{f}_t = f(x_t^{(z)}, c) - \hat{f}(x_t^{(z)}, \hat{c})$. Note that equation 33 is a second-order inhomogeneous linear differential equation.

1) *General solution:* Making the substitution in the differential equation, $r$ satisfies the auxiliary equation:

$$r^2 - \frac{K_v}{m} = 0 \Rightarrow r_{1,2} = \pm\sqrt{\frac{K_v}{m}} \tag{34}$$

then, we obtain the general solution of the differential function $\bar{x}_e$ as:

$$\bar{x}_e = C_1 e^{r_1 x} + C_2 e^{r_2 x} \tag{35}$$

2) *Special solution:* Consider the standard second order differential equation: $\ddot{y} + p\dot{y} + qy = P(x)e^{\alpha x}$, the special solution $x_e^*$ is obvious:

$$x_e^* = -\epsilon/K_v \tag{36}$$

Therefore, $x_e$ can be expressed as:

$$x_e = \bar{x}_e + x_e^* = C_1 e^{r_1 x} + C_2 e^{r_2 x} - \epsilon/K_v \tag{37}$$

3) *Calculation of $C_1$ and $C_2$:* In the equation 37, there exist two fixed points:$(x_d,0)$ and $(0,-x_d)$. We have:

$$\begin{cases} 0 = C_1 e^{r_1 x} + C_2 e^{r_2 x} - \epsilon/K_v \\ -x_d = C_1 + C_2 - \epsilon/K_v \end{cases} \tag{38}$$

By solving the simultaneous formulas, we will get the answer of $C_1$ and $C_2$.

$$\begin{cases} C_1 = -x_d - C_2 + \epsilon/K_v \\ C_2 = (\epsilon + (K_v x_d - \epsilon)e^{r_1 x_d})/K_v(e^{r_2 x_d} - e^{r_1 x_d}) \end{cases} \tag{39}$$

Thus the solution of the original derivative function is:

$$x_e = x_t^{(z)} - x_t^d = C_1 e^{r_1 x} + C_2 e^{r_2 x} - \epsilon/K_v$$

and we have the average error bound in equation 2:

$$
\begin{aligned}
ACE_{\pi_\delta} &= \frac{1}{T}\sum_{t=1}^{T}\|\mathbb{E}_{z\sim\pi_\delta}(x_t^{(z)} - x_t^d)\| \\
&= \frac{1}{T}\sum_{t=1}^{T}\|\mathbb{E}_{z\sim\pi_\delta}[C_1 e^{r_1 x_t^{(z)}} + C_2 e^{r_2 x_t^{(z)}} - \epsilon/K_v]\| \\
&= \frac{1}{T}\sum_{t=1}^{T}\|\mathbb{E}_{z\sim\pi_\delta}[C_1 e^{r_1 x_t^{(z)}} + C_2 e^{r_2 x_t^{(z)}}] - \mathbb{E}_{z\sim\pi_\delta}(\epsilon/K_v)\| \\
&= \frac{1}{T}\sum_{t=1}^{T}\|\mathbb{E}_{z\sim\pi_\delta}[C_1 e^{r_1 x_t^{(z)}} + C_2 e^{r_2 x_t^{(z)}}] - \mathbb{E}_{z\sim\pi_\delta}[f(x_t^{(z)}, c) - \hat{f}(x_t^{(z)}, \hat{c})]/K_v\| \\
&= \frac{1}{T}\sum_{t=1}^{T}\|\mathbb{E}_{z\sim\pi_\delta}[C_1 e^{r_1 x_t^{(z)}} + C_2 e^{r_2 x_t^{(z)}}] - \tilde{h}^{-1}(p)/K_v\|
\end{aligned}
\tag{40}
$$

## B  EXPERIMENTAL SETTINGS AND DETAILS

### B.1  OOD-CONTROL PSEUDO CODE AND ALGORITHM SETTINGS

Out-of-Distribution comes from the misspecification of systems' components and the systematic error of sensors and environment models. Our algorithm is able to extrapolate unknown wind disturbances after learning a model from previous data containing generalized information.

To eliminate the influence caused by other factors, the two models share the same initial state and environmental conditions. Furthermore, in order to make the training process fair for both models, we simulate them for the same number of iterations and sustain each iteration for the same period of time.

---

**Algorithm 1** OoD-Control (Out-of-Distribution Generalization control for Adaptive Nonlinear Control)

---

**Input**: Set of distribution functions X , DNN $\phi$ with parameter $\Theta$, environment estimation vector $\hat{c}$
**Parameter**: Parameters of mechanical system and aerodynamics.
**Output**: The estimation of unknown force $\hat{f}$

1: **while** picking $\chi$ from X **do**
2:     Sample a series of independent random variables $w$ subject to $\chi$ as external wind force.
3:     Apply external force to the simulation according to equation 43
4:     Calculate loss with noise in the state: $L = ||\phi(x + \Delta x)^T \hat{c} - f||^2$
5:     Update $\Theta : \Theta = \Theta - \eta_1 \nabla_\theta L$
6:     Update $\hat{c} : \hat{c} = \hat{c} - \eta_2 \nabla_{\hat{c}} L$
7:     **return** $\hat{f} = \phi(x; \Theta)^T \hat{c}$
8: **end while**

---

### B.2  UPDATE OF PARAMETERS $\phi$ AND $\hat{c}$

Based on equation 1, we design a discrete-time simulation process to calculate state variables and estimate the unknown term at each time interval and by this way, collect data for training $\phi$. In algorithm 1, we update $\Theta$ during simulation: $\Theta = \Theta - \nabla L_\theta$.

Keeping a constant $\hat{c}$ in the inverted pendulum experiment results in the inability to update environmental parameters. This can result in higher ACE or even failure to control the system (see Table 2). For the quadrotor, when the wind is severe, it cannot maintain its position, as illustrated in Figure 3. It should be noted, however, that our algorithm exhibits better control even when position drift occurs. Generally, updates to $\phi$ would be more energy intensive, whereas updating only $\hat{c}$ would be closer to the actual embedded device.

Table 1: ACE results in pendulum experiments with the changed or unchanged $\hat{c}$

| $\hat{c}$ | Test Env. | No-Adapt | OMAC | OoD-Control | Omniscient |
|---|---|---|---|---|---|
| Changed | Breeze | 0.538(0.325) | 0.054(0.022) | 0.050(0.024) | 0.046(0.024) |
| Changed | Strong Breeze | 1.566(0.257) | 0.134(0.032) | **0.078(0.029)** | 0.046(0.024) |
| Changed | Gale | 2.277(0.427) | 0.592(0.196) | **0.163(0.055)** | 0.045(0.021) |
| Unchanged | Strong Breeze | 1.566(0.257) | 1.553(0.262) | 1.548(0.265) | 0.046(0.024) |
| Unchanged | Gale | 2.277(0.427) | 2.276(0.425) | 2.275(0.425) | 0.045(0.021) |

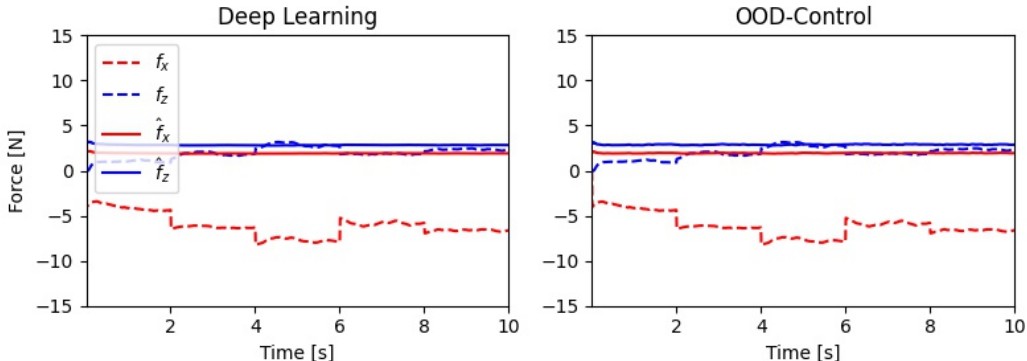

Figure 3: Results when both $\hat{c}$ and $\phi$ keep unchanged in testing

### B.3 Pseudo code of the perturbation radius calculation algorithm

---

**Algorithm 2** Monte Carlo algorithm for calculation of perturbation radius

---

**Input**: $x$, number of Monte Carlo samples: N, the variance of Gaussian noise: $\sigma$
**Parameter**: threshold of error: $\epsilon$
**Output**: radius

1: **for** i in range(N) **do**
2:     Add Gaussian noise $x = x + \mathcal{N}(0, \sigma)$ and append to x_set
3: **end for**
4: **for** x_sample in x_set **do**
5:     Obtain the prediction value and calculate prediction error
6:     **if** error $< \epsilon$ **then**
7:         nA = nA+1     // nA is the number of successful predictions
8:     **end if**
9: **end for**
10: pABar ← calculate the lower confidence bound
11: **if** pABar $< 0.5$ **then**
12:     **return** 0    // 0 means abstention
13: **else**
14:     radius = $\sigma \cdot f_{ppa}(\text{pABar})$    // $f_{ppa}$ is the percent point function of Gaussian distribution.
15: **end if**
16: **return** radius

---

### B.4 Differential equation solver

The fourth-order Runge-Kutta method is a common iterative method to calculate differential equations and approach continuous functions. In this paper, we use this method to calculate equation 1 and simulate the process of the mechanical system. We do the simulation by moving tiny $\Delta t$ each

time, which corresponds to the step size of the Runge-Kutta method. And the right-hand side of equation 1 is the derivative of the state $x$.

## B.5 THE COMPARISON OF VARIOUS TRAJECTORY TRACKING MODES

OoD generalized model adapts better and gives inputs forces closer to the desired ones as illustrated in Figure 4 to Figure 7 which shows the traces under OMAC (Deep Learning), OoD-Control and omniscient models and the desired trajectory, the OoD-Control algorithm clearly goes closer to the desired curve.

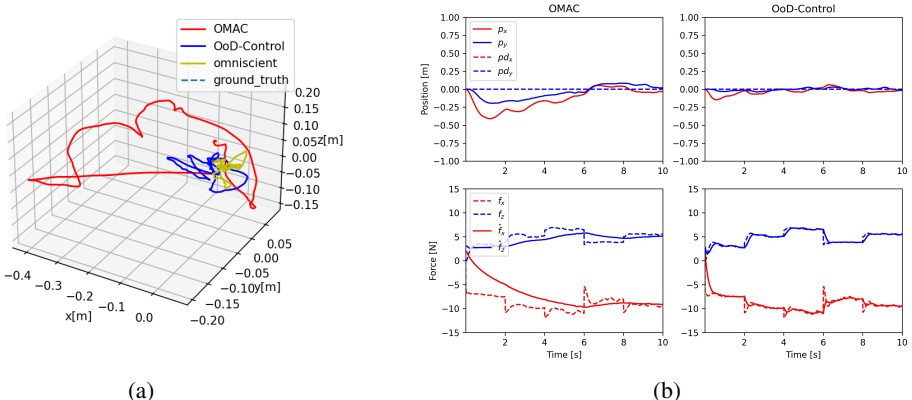

Figure 4: Traces when the quadrotor tries to keep still.

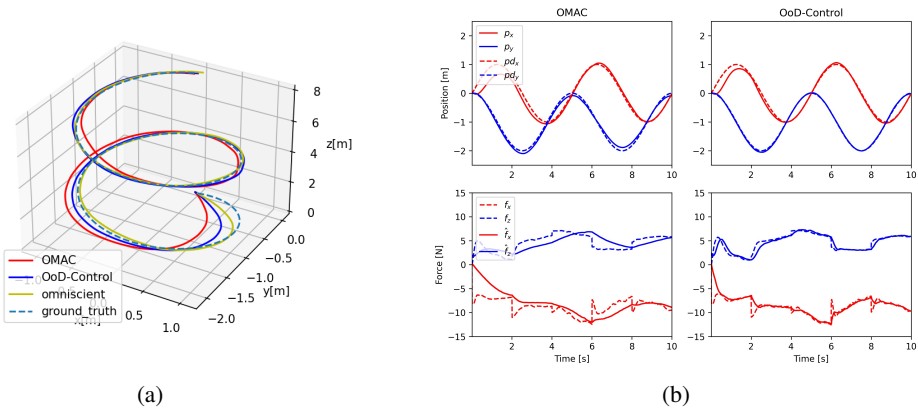

Figure 5: Traces when the trajectory is a spiral curve.
.

## B.6 WIND CONDITIONS IN EXPERIMENTS

### B.6.1 WIND FIELD SETTINGS

**Protocol** Two sets of distribution functions $X$ and $\Omega$ with $X \cap \Omega = \emptyset$ are defined for training and testing. The training distribution $\chi \in X$ and the testing distribution $\omega \in \Omega$ are specified for different experiment tasks. The wind velocity is a series of independent random variables sampled subject to the distributions ($\chi$ or $\omega$) picked out. The wind brings induced airflow to the rotor blades, creating complex and nonstationary aerodynamic interactions[2]. All models are simulated with the same wind series and the simulating duration is also the same.

---

[2]More detailed information related to the aerodynamics under wind is shown in Appendix B.6.3

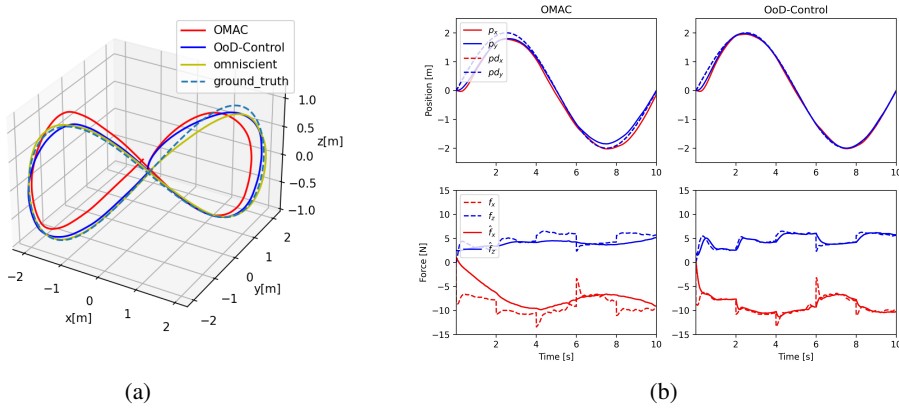

Figure 6: Traces of OMAC, OoD-Control and omniscient quadrotor models with figure-8 trajectory

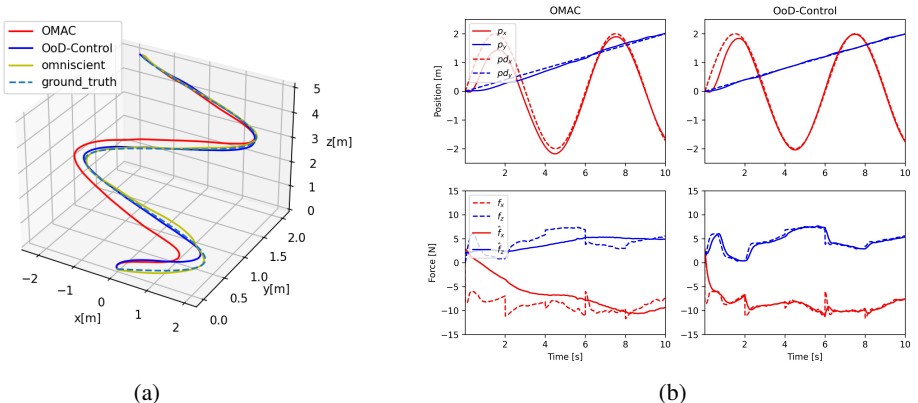

Figure 7: Traces of OMAC, OoD-Control and omniscient quadrotor models with sin-forward trajectory.

Table 2: ACE in quadrotor experiment with different trajectories and environment

| Trajectory | Test Env. | No-adapt | OMAC | OoD-Control | Omniscient |
|---|---|---|---|---|---|
| hover | Breeze | 0.279(0.056) | 0.202(0.035) | **0.048(0.007)** | 0.029(0.003) |
| hover | Strong Breeze | 0.410(0.085) | 0.257(0.049) | **0.060(0.011)** | 0.035(0.003) |
| hover | Gale | 0.478(0.100) | 0.297(0.079) | **0.074(0.026)** | 0.040(0.004) |
| sin-forward | Breeze | 0.319(0.058) | 0.277(0.047) | **0.098(0.011)** | 0.079(0.010) |
| sin-forward | Strong Breeze | 0.439(0.086) | 0.334(0.053) | **0.110(0.017)** | 0.085(0.016) |
| sin-forward | Gale | 0.504(0.101) | 0.379(0.067) | **0.133(0.045)** | 0.092(0.021) |
| figure-8 | Breeze | 0.334(0.060) | 0.268(0.032) | **0.094(0.005)** | 0.079(0.005) |
| figure-8 | Strong Breeze | 0.455(0.085) | 0.318(0.055) | **0.100(0.006)** | 0.082(0.005) |
| figure-8 | Gale | 0.520(0.100) | 0.403(0.216) | **0.169(0.195)** | 0.085(0.006) |
| spiral-up | Breeze | 0.295(0.060) | 0.214(0.044) | **0.084(0.012)** | 0.067(0.004) |
| spiral-up | Strong Breeze | 0.422(0.088) | 0.265(0.057) | **0.095(0.016)** | 0.071(0.006) |
| spiral-up | Gale | 0.490(0.103) | 0.309(0.090) | **0.120(0.055)** | 0.075(0.008) |

**Pendulum** In this task, we use 5 different normal distributions as the training set, i.e. $X = \{\chi_k | \chi_k = N(0, 0.2k), k = 1, 2, \ldots, 5\}$. When testing, we set 3 different levels of the wind based on the difficulty, that is the breeze, strong breeze, and gale. And each level has 6 or 8 different uniform distributions. We show the details of the testing set $\Omega$ as follows:

- Breeze: $\Omega_1 = \{\omega_1^k \mid \omega_1^k = U(-0.5k, 0.5k), k = 1, 2, \ldots, 5, 6\}$
- Strong Breeze: $\Omega_2 = \{\omega_2^k \mid \omega_2^k = U(-3.0 - 0.5k, 3.0 + 0.5k), k = 1, 2, \ldots, 5, 6\}$
- Gale: $\Omega_3 = \{\omega_3^k \mid \omega_3^k = U(-6.0 - 0.5k, 6.0 + 0.5k), k = 1, 2, \ldots, 7, 8\}$

**Quadrotor** In this instantiation, we only use one training distribution, the three-dimensional standard normal distribution, to train the models. Meanwhile, we transform the initial wind data sampled from the normal distribution to their absolute values, i.e. $\chi = |N(\mu, \Sigma)|, \mu = [0, 0, 0], \Sigma = diag(1, 1, 1)$. The reason for this operation is that we want to make the wind come from only one octant, then it differs from the test environment in distribution and direction. Similarly, the test set $\Omega$ also has 3 different levels, but the exact distributions are different from the pendulum task. Following are the details:

- Breeze: $\omega_1 = U(D_1), D_1 = \{(x, y, z) \mid x, y, z \in (-3, 3)\}$
- Strong Breeze: $\omega_2 = U(D_2), D_2 = \{(x, y, z) \mid x, y, z \in (-6, 6)\}$
- Gale: $\omega_3 = U(D_3), D_3 = \{(x, y, z) \mid x, y, z \in (-8, 8)\}$

**Trajectory illustration** The three trajectories we used in the quadrotor experiment (see Table 2) are mathematically described as:

- sin-forward: $(x, y, z) = (2sin(\frac{\pi t}{3}), 0.2t, 0.5t)$
- figure-8: $(x, y, z) = (2sin(\frac{\pi t}{5}), 2sin(\frac{\pi t}{5}), sin(\frac{2\pi t}{5}))$
- spiral-up: $(x, y, z) = (sin(\frac{2\pi t}{5}), cos(\frac{2\pi t}{5}) - 1, 0.2t)$

### B.6.2 DISCUSSION

**Q1:** *What's the main problem addressed in this paper and what's the innovation compared with other online adaptive control?*

**A1:** We study the online adaptive flight control problem when testing and training environment domain shifts and give a methodology that the upper bound of the predicted error of the unknown dynamics maintains constant under a radius of perturbation.

**Q2:** *How does the OoD-Control algorithm perform in the i.i.d. environment, where the train and test environment follow the same distribution?*

**A2:** We run OMAC and OoD-Control algorithm on i.i.d. environment and put the results in Table 3 (see Appendix B.7). In the i.i.d. environment, the ACE of all algorithms drop naturally, but OoD-Control still performs much better than OMAC.

**Q3:** *How does the size of noise add to the state vectors when training affects the performance of our algorithm?*

**A3:** We run OoD-Control algorithm with different noises in i.i.d. and o.o.d. environment respectively. Results (see Table 4) show that the algorithm performs worse in the i.i.d. environment as the scale of noise increases. And performance in the OoD environment gets a little better in a certain range when noise size increases, but becomes worse when noise is too large.

### B.6.3 AERODYNAMICS OF THE ROTORS IN WIND

Wind conditions affect the quadrotor mainly in two ways. The first is that it alters the aerodynamics of the rotors. The second is to change the air resistance of each windward side. The aerodynamics of the rotor in windy conditions is illustrated in Figure 8. This paper defines several types of winds, namely Breeze, Strong Breeze, and Gale.

Table 3: ACE in quadrotor experiment in i.i.d. environment

| Trajectory | No-adapt | OMAC | OoD-Control | Omniscient |
|---|---|---|---|---|
| hover | 0.378(0.052) | 0.138(0.030) | **0.036(0.012)** | 0.027(0.003) |
| figure-8 | 0.436(0.043) | 0.230(0.030) | **0.087(0.008)** | 0.075(0.005) |
| spiral-up | 0.403(0.050) | 0.158(0.031) | **0.076(0.008)** | 0.065(0.003) |
| sin-forward | 0.432(0.048) | 0.209(0.022) | **0.092(0.017)** | 0.076(0.004) |

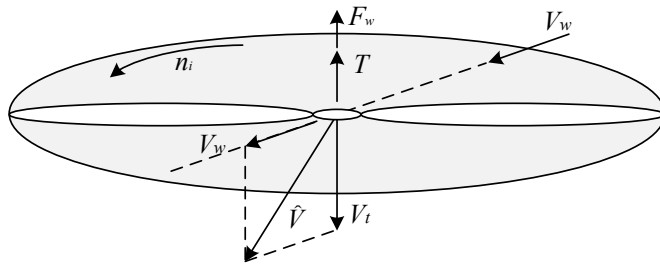

Figure 8: Sketch of the aerodynamics of the propeller in wind conditions

1) *Aerodynamics caused by wind gust disturbances.* According to the rotor slipstream theory (Conlisk, 2001), the induced velocity is calculated by:

$$\|V_t\| = \sqrt{\frac{k_f n^2}{2\pi\rho r^2}} \tag{41}$$

where $k_f$ is the lift coefficient, $\rho$ is the air density, $r$ is the rotor radius. A wind field results in a total aerodynamic force on the rotor equal to the sum of the lift force $T$ and the additional wind disturbance force $F_w$. The total lift can be calculated as:

$$\|T + F_w\| = 2\pi\rho r^2 \|V_t\| \|V_w + V_t\| \tag{42}$$

Therefore, the wind disturbance force $\mathcal{H}_{w_i}$ and moment $\mathcal{M}_{w_i}$ on $i_{th}$ rotor is:

$$\begin{cases} \mathcal{H}_{w_i} = k_f n_i^2 - 2\pi\rho r^2 \|(0, 0, \sqrt{k_f n_i^2/2\pi\rho r^2})^T + V_w^B\| \\ \mathcal{M}_{w_i} = \begin{cases} k_m \mathcal{H}_{w_i}/k_f, \text{ when the rotor turns clockwise} \\ -k_m \mathcal{H}_{w_i}/k_f, \text{ otherwise} \end{cases} \end{cases} \tag{43}$$

where $k_m$ is the anti-torque coefficient related to the shape of the rotors and local air density.

2) *The air drag.* Air drag can be ignored in hovering or low-speed flights without wind. However, in the presence of a wind field, the following equation can be used to calculate air drag.

$$D_g = \frac{1}{2} c\rho S_{air} V_{air}^2 \tag{44}$$

where $c$ represents the air drag coefficient, $S_{air}$ is the windward area, and $V_{air}$ is the relative speed of the wind to the quadrotor.

### B.7 PERFORMANCE IN I.I.D. ENVIRONMENT AND DIFFERENT NOISE SETTING

$Noise_x$ and $Noise_a$ mean the noise scale added to the state vector and the environment representation vector respectively. In this experiment, We double the test duration (compared to the results in Table 2) to enlarge the differences.

Table 4: ACE in quadrotor experiment with different noise

| Trajectory | OoD Env. | Noise$_x$ | Noise$_a$ | i.i.d. | o.o.d. |
|------------|----------|-----------|-----------|--------|--------|
| figure-8 | Gale | 0.01 | 0.01 | 0.0628 | 0.5897 |
| figure-8 | Gale | 0.02 | 0.02 | 0.0635 | 0.5897 |
| figure-8 | Gale | 0.05 | 0.02 | 0.0652 | 0.5767 |
| figure-8 | Gale | 0.05 | 0.05 | 0.0651 | 0.5877 |
| figure-8 | Gale | 0.1 | 0.1 | 0.0729 | 0.5908 |

