# OpenReview forum: "OoD-Control: Out-of-Distribution Generalization for Adaptive UAV Flight Control"
_ICLR.cc/2023/Conference — Submitted to ICLR 2023_

### Official Review · Reviewer_ma1x · 2022-10-24

**Confidence:** 4
**Correctness:** 2
**Technical Novelty And Significance:** 2
**Empirical Novelty And Significance:** 2
**Recommendation:** 3

**Clarity, Quality, Novelty And Reproducibility:**

Novelty is a concern since it is not clear how exactly the out-of-distribution data is addressed. A high-level explanation or description of the approach is also missing from the paper.

**Strength And Weaknesses:**

Strength:

+ Combining strengths of data-driven methods and control methods is attractive, especially for robotics applications such as controlling UAVs.

+ The control approach is mathematically formulated and analyzed.

Weakness:

- It seems the argued novelty focuses on addressing out-of-distribution data. However, the paper simply uses a deep neural network (inspired by Shi et al. 2021) to model the unknown environment-dependent dynamics. Why is this neural network capable of modeling such an unknown dynamic during the execution time? How does this neural network model the environment? If the dynamics in the execution phase is unknown and different from the training phase, how to especially model or address this difference?

- The integration of data-driven methods and control theories is simple. The proposed approach assumes the expectation of the prediction of the unknown dynamics remains the same for a perturbation of radius, and uses conventional control theories to analyze the bound. However, this assumption may not be true in the real-world scenario. Even if it is true, how to deal with the transition between areas that may have different perturbations.

- What is the relationship between out-of-distribution data and perturbations?

- When comparing with previous out-of-distribution generalization methods in related work, the paper argues previous methods suffers from “neglecting the gap between the simulation experiments and reality”. The proposed method is also not evaluated on a real system. It is not convincing that the proposed method can be applied to a real UAV.

- The paper generally argues that it is difficult or impractical to model the environment-dependent dynamics. Then in the experiment, the paper uses the widely used Dryden model to simulate wind field. Given this Dryden model, why is it challenging to model the dynamics?


**Summary Of The Paper:**

This paper describes a data-driven control approach that is argued to address the out-of-distribution control problem. The approach models unknown environment-dependent dynamics of a UAV using a deep neural network, and then use conventional control theories for UAV controls.

**Summary Of The Review:**

Although the idea of integrating control theories with learning methods is attractive, novelty of the paper is a concern and applicability of the approach to real UAVs is also a concern.

---

> ### Author Response · Authors · 2022-11-12
> **Reply to Reviewer ma1x for previous review**
>
> #### **Q1: Why is this neural network capable of modelling such an unknown dynamic during the execution time? How does this neural network model the environment? If the dynamics in the execution phase are unknown and different from the training phase, how to especially model or address this difference?**
>
> **A1:** In the *Notation and Settings* paragraph (Section 4), we have mentioned that the unknown environment-dependent dynamics typically consist of two coupling parts: the irregular higher-order aerodynamics of UAVs that are caused by the complex streamlined design, and the environment-dependent variables which encode wind field information. Under these settings, the deep neural network (DNN) is used to model the high-order part of the unknown dynamics, rather than the whole dynamics. We actually get inspiration from Shi et al.,(2021) that a DNN would lead to a highly non-convex learning object. If the controller can learn the environment-invariant representations, then it can perform much better when the distribution shifts.
>
> #### **Q2: The proposed approach assumes the expectation of the prediction of the unknown dynamics remains the same for a perturbation of radius and uses conventional control theories to analyze the bound. However, this assumption may not be true in a real-world scenario. Even if it is true, how to deal with the transition between areas that may have different perturbations?**
>
> **A2:** Thanks for pointing out the potential problem of the mentioned assumption in the real world. But it seems that it is not the exact assumption we made in our proof. It is actually what we aim to prove. The perturbation with radius $r$ is used for training the model to be more capable of generalization, not the one relative to the environment. Maybe it is our confusing expression in *Section 4.3* that makes you misunderstand. We have modified this problem in the latest revision.
>
> #### **Q3: What is the relationship between out-of-distribution data and perturbations?**
>
> **A3:** The out-of-distribution (OoD) data are related to our task settings. We aim to find a robust controller that can still predict the unknown dynamics well when the wind distribution has shifted. The out-of-distribution property belongs to wind data (or environments).
> The perturbations are related to our methodology. We add perturbations to the input of our DNN in order to improve its robustness. And we place the perturbations in our certifiable proof to verify the bound guarantee of our method.
>
> #### **Q4: In the experiment, the paper uses the widely used Dryden model to simulate wind field. Given this Dryden model, why is it challenging to model the dynamics?**
>
> **A4:** The Dryden model is a widely used statistical model to approximate the dynamics. Given the fact that a small UAV is unable to carry heavy wind detectors, it is unlikely for it to get accurate wind conditions online. We use the Dryden model as an example to demonstrate how to tackle unseen environments.
>
> In our experiments, even if we only use the Dryden model to simulate the wind field, it's still the fact that we make the testing environment distributions different from the training distributions. That makes the testing environment "unseen" for the UAV, which is a challenging task. And the experiment results show that our method performs much better than the OMAC.
>
> #### **Q5: Novelty is not clear and a high-level explanation of the approach is missing.**
>
> **A5:** As *Section 1* (Introduction) claimed, we propose an expandable algorithm to improve UAV control performance in unseen environments. Our method does not require strong assumptions and guarantees a bound of control error under domain shifts of environment distribution.
>
> We add random noise w.r.t. wind condition and state to state vectors in training, so a generalized distribution of state is learned under the same environment.

---

### Official Review · Reviewer_r1g3 · 2022-10-24

**Confidence:** 3
**Correctness:** 2
**Technical Novelty And Significance:** 3
**Empirical Novelty And Significance:** 3
**Recommendation:** 5

**Clarity, Quality, Novelty And Reproducibility:**

The paper's clarity could be improved. The authors must still convince the reviewer that all their math is correct (see above). The paper's conclusions are an extension of Shi et al. (2021)'s work and seem novel to the reviewer. For reproducibility, the authors should make the text (and appendix) more self-contained.

**Strength And Weaknesses:**

**Strength**:

The problem is very relevant and the experimental results look very good.

**Weaknesses:**

The paper follows Shi et al. (2021) very closely and does not always explain the terminology and choices. This makes the paper sometimes hard to read, as the theoretical contribution comes across as somewhat unmotivated and require large jumps in reasoning. The problem is compounded by some (in this reviewer's opinion) apparent errors in the math. More thoroughness and more explanation on the assumptions, used transformations and definitions would probably improve the paper a lot.
The reviewer would like the authors to explain the following:
1. Is the defined ACE measure useful for OoD generalization? Note that $|| \mathbb E[x_t^{(z)}] - x_t^d ||$ is zero for symmetric effects, e.g. wind blows equally often from left and right. A controller with ACE=0 can therefore still perform arbitrarily bad! Shi et al. (2021) seem to optimize $\mathbb E[||x_t^{(z)} - x_t^d ||]$. Please compare the two definitions and state your reasoning for choosing one.
2. Section 4.1 introduces an important *required* property; how does Th.1 prove this?	Assuming the right-hand side of (eq.5) is $p$, how do you guarantee that $\tilde h_{\pi_0}(x_t^{(z)}) > p$?
3. Theorem 1 uses the substitution $h_{\pi_0}(x_t^{(z)} + \delta)$ to $h_{\pi_\delta}(x_t^{(z)})$, where $\pi_{\delta}$ is defined as "the distribution of $z + \delta$". How does the transformation $x_t^{(z)} + \delta$ to $x_t^{(z+\delta)}$ work? Don't these $\delta$'s have different dimensionality?
4. The current use of $\tilde h$ is very unintuitive. Could you elaborate how one has to interpret $\tilde h_{\pi_0}(x_t^{(z)})$, in particular for the crucial (eq.8)?
5. (eq.10) and (eq.11) seem to ignore $x_{t+1}^d$. To make sense in (eq.12), (eq.10) should have $x_{t+1}^d$ in the RHS bracket and the LHS of (eq.11) should be $x_{t+1}^{(z)} - x_{t+1}^d$.  Similarly, Corollary 2 seems to miss $x_{t+1}^d$ in the definition of $e_f$.
6. The $p$ in (eq.12) is not defined, but combining the reviewers interpretation of (eq.8) and (eq.11) would mean $p$ is the rhs of (eq.8). Is this correct?
7. How important is Corollary 2? Can you make an ablation with and without considering control misspecification? Also, where does $K_v$ come from?
8. Algorithm 1 is not understandable without reading Shi et al. (2021).	Please augment the appendix with a self-contained description.
9. Please indicate which experimental results are statistically significant. For example, in Table 1 only row 3 (and maybe 2) look as if OoD is significantly better than OMAC, whereas in Table 2 all **but** row 9 look significant. Just claiming "is higher" is not meaningful.


**Summary Of The Paper:**

The paper aims for out-of-distribution adaptation of UAV controllers, e.g., in the presence of unexpected wind. The authors derive an upper bound on their version of the average control error and an algorithm (OoD-Control) to adapt the dynamics model from samples of the changed dynamics. Experiments on the inverted pendulum and UAV control show improved performance in comparison to OMAC.


**Summary Of The Review:**

The paper presents an interesting and promising approach to an important problem. The text is sometimes hard to read, and the theory might have some errors (see questions). However, the reviewer is willing to improve the score if the authors can improve readability and convince me that the math is correct (or fix it).

---

> ### Author Response · Authors · 2022-11-12
> **Reply to Reviewer r1g3 for previous review (Part 1)**
>
> #### **Q1: Is the defined ACE measure useful for OoD generalization? Please compare the two definitions and state your reasoning for choosing one.**
>
> **A1:** For quadrotors, if the wind is equal on the left side and the right side, it can be regarded as a windless environment. ACE represents the difference between the quadrotor's current and ideal state. Taking the fixed-point hovering scenario as an example, ACE=0 indicates that the quadrotor can be precisely positioned at the ideal point. If the quadrotor is unable to hover or is unstable, ACE will be a very large value rather than 0.
>
> In Shi et al.(2021), the average control error (ACE) is defined as $ACE=\frac{1}{TN}\sum_{i=1}^N\sum_{t=1}^T\|x_t^{(i)}\|$. For the OoD problem, we introduce a perturbation $z$ that follows a certain distribution in our definition. Our definition of ACE differs from that of Shi et al.(2021) in that $x$ computes the expectation under different perturbations. And the definition of Shi et al.(2021) cannot be valid in the case of the OoD generalization.
>
> #### **Q2: Section 4.1 introduces an important required property; how does Th.1 prove this? Assuming the right-hand side of (eq.5) is $p$, how do you guarantee that $\tilde{h}_{\pi 0}(x_t^{(z)})>p$?**
>
> **A2:** In *Section 4.1*, we want to prove that $\forall \|\delta\|\leq r$, $\exists{p > 0}$, $\tilde{h}_{\pi_0}(x_t^{(z)})>p$, it still holds $\tilde{h}\_{\pi\_0}(x^{(z)}\_t+{\delta}) > p$. This means that for a perturbation of radius $r$, the expectation of the unknown dynamics remains constant.
>
> #### **Q3: How does the transformation $x_t^{(z)} + \delta$ to $x_t^{(z+\delta)}$ work? Don't these $\delta$s have different dimensionality?**
>
> **A3:** Given a sequence of $x$ in $t$: $x_t=(x_{1,t},x_{2,t},\dots,x_{n,t})$. And the $z$ sequence is $z=(z_1,z_2,\cdots,z_n)$. Then $x_t^z=x_t+z=(x_{1,t}+z_1,x_{2,t}+z_2,\dots,x_{n,t}+z_n)$. The perturbation $\delta$ sequence is $\delta=(\delta_1,\delta_2,\dots,\delta_n)$. $x_t^z+\delta =(x_{1,t}+z_1+\delta_1,x_{2,t}+z_2+\delta_2,\dots,x_{n,t}+z_n+\delta_n)$. (In fact, we do not use the notation $x_t^{(z+\delta)}$ in the paper.) To be additive, $\delta$ and $x$ have the same dimensions.
>
> Thus the dimension of $\delta$ is $n$ and the set of the perturbation is $\mathcal{B}=\{{{\delta} \in R^{n}:\|{\delta}\|_2\leq r}\}$ . The correction of $\delta$ is also updated in the revised version of the paper.
>
> #### **Q4: Could you elaborate how one has to interpret $\tilde{h}_{\pi_0}(x_t^{(z)})$, in particular for the crucial (eq.8)?**
>
> **A4:** We perform several transformations of the state $x^z_t$ of the system in the mathematical proof. In this paper, $\tilde{h}_{\pi_0}(x^{(z)}_t)$ is the shorthand for $\tilde{h}(|f\_{\pi\_0}(x^{(z)}\_t)-\hat{f}\_{\pi\_0}(x^{(z)}\_t)|)$ (see A5 for reviewer 7hFa). $x_t^z$ is the state vector under perturbance $z$ and $f\_{\pi\_0}(x^{(z)}\_t)={E}\_{{z}\sim \pi\_0}[f(x^{(z)}\_t,c)]$. Then we use $\tilde{h}$ to map the loss $\ell\_f=|f\_{\pi\_0}(x^{(z)}\_t)-\hat{f}\_{\pi\_0}(x^{(z)}\_t)|$ to $[0,1]$.
>
> #### **Q5: (eq.10) and (eq.11) seem to ignore $x_{t+1}^d$. To make sense in (eq.12), (eq.10) should have $x_{t+1}^d$ in the RHS bracket and the LHS of (eq.11) should be $x_{t+1}^{(z)}-x_{t+1}^d$. Similarly, Corollary 2 seems to miss $x_{t+1}^d$ in the definition of $e_f$.**
>
> **A5:** The (eq.10) is a controller designed in the ideal case, and for the convenience of representation, here we consider the scenario of fixed-point hovering in *Lemma 1* and *Corollary 2* and the desired point $x^d_{(t+1)}$ is $(0,0,0)$. The trajectory tracking ACE is in *Lemma 2*.
>
> #### **Q6: The $p$ in (eq.12) is not defined, but combining the reviewer's interpretation of (eq.8) and (eq.11) would mean $p$ is the RHS of (eq.8). Is this correct?**
>
> **A6:** The $p$ is defined as the lower bound of $\tilde{h}(x^{(z)}_t)$ in *Section 4.1*. And in (eq.8), the RHS is the value of $\tilde{h}(x^{(z)}_t)$, the lower bound of which is $p$ by definition.
>
> #### **Q7: How important is Corollary 2? Can you make an ablation with and without considering control misspecification? Also, where do $K_v$ come from?**
>
> **A7:**
> 1. In the real world, the parameters of physical models are modelled with some uncertainty. Also, differences in modelling may cause instability in the sys modelling we discuss the case where there is misspecification in the controller model parameters. This is also the motivation for *Corollary 2*.
> 2. $K_v$ is the weight matrix of the trajectory tracking error, which is used to tune the magnitude of the desired control force. We missed the description of $K_v$ in the article and will add it to the revised version.
>
> #### **Q8: Algorithm 1 is not understandable without reading Shi et al.(2021). Please augment the appendix with a self-contained description.**
>
> **A8:** Thank you very much for pointing out this problem. We have augmented the appendix in the latest revision.

---

> > ### Comment · Reviewer_r1g3 · 2022-11-12
> > **Thanks for the answers**
> >
> > Thanks to the authors for such a detailed answer! I still believe that a major rewrite of the paper, which explains the main story and the place of each theoretical statement in it better, would improve the papers readability a lot. However, the author's answers to my questions have removed most theoretical questions I had (except Q2 and Q6!), although I still do not see the big picture of the paper, and the role of each lemma, corollary and theorem in it. If I would be the only reviewer this would be enough to reject the paper, but as I am not as familiar with comparable literature as other reviewers, I will follow their lead in the decision making process. In the following I comment on the author's answers to my questions:
> >
> > 1. I agree that for planning an average trajectory a model that describes the average outcome (under disturbance) is sufficient, but one could also aim for a trajectory where *each likely disturbance* reaches the destination (akin to robust planning), in which case a different definition of the ACE would be better. Restricting yourself to the first case is fine, but should be mentioned in the paper.
> > 2. This does not answer my question. I understand why the *required property* is useful, but I do not see how Theorem 1 proves the property holds. If it does not, why is this subsection and the Theorem (and Corollary) here? Please explain.
> > 3. That answered my question, thank you.
> > 4. Please add this shorthand to the text. Also, please add a remark that a lower bound on h is an upper bound on the error.
> > 5. Does the paper actually specify that the equation is for a special case (ideal case does to the best of my knowledge not specify to target the origin)? Please add the targets, otherwise readers will be confused. Also specify the pseudo-inverse symbol (dagger).
> > 6. This is very confusing. You state that the lower bound $\tilde h_{\pi_0}(x_t^z) > p$ exists. I am not convinced of that because of Q2 above. However, if it does, why should the *same* bound hold for $\pi_\delta$? In the proof of Lemma 1, where did $\mathbb E[||w_t||]$ go?
> > 7. If I understand you correctly, the misspecification $\Delta B$ is unknown and Corollary 2 shows that the ACE under misspecification can be bounded (with the additional term). Please say that directly.
> > 8. I'm afraid this is still not understandable enough. What are $L_\theta$ and $L_{\hat c}$? Are these different losses? Where are they defined? Or does $\nabla L_\theta$ denote the gradient of loss $L$ w.r.t. $\theta$? In the latter case, please use $\nabla_\theta L$.
> > 9. Please clarify in the table description how significant results are marked (I assume it is bold, but having check-marks everywhere does not improve reading). Please indicate statistical significance in Tables 2 and 3!

---

> > > ### Author Response · Authors · 2022-11-17
> > > **Reply for the precious addtion comments**
> > >
> > > #### **Q1: I agree that for planning an average trajectory a model that describes the average outcome (under disturbance) is sufficient, but one could also aim for a trajectory where each likely disturbance reaches the destination (akin to robust planning), in which case a different definition of the ACE would be better. Restricting yourself to the first case is fine, but should be mentioned in the paper.**
> > >
> > > **A1:** Thanks for your comments and suggestions. We have added a description of the application scenarios in our paper.
> > >
> > > #### **Q2: This does not answer my question. I understand why the required property is useful, but I do not see how Theorem 1 proves the property holds. If it does not, why is this subsection and the Theorem (and Corollary) here? Please explain.**
> > >
> > > **A2:** I appreciate you asking this question. In Corollary 1, we demonstrate the detailed proof procedure for the case where p = 0.50. See *Appendix A.2* for details.
> > >
> > > #### **Q3: That answered my question, thank you.**
> > >
> > > #### **Q4: Please add this shorthand to the text. Also, please add a remark that a lower bound on h is an upper bound on the error.**
> > >
> > > **A4:** Thank you very much for your comments and suggestions. We have added a description of the application scenarios in our paper. The shorthand of $\tilde{h}(\|f\_{\pi\_0}(x^{(z)}\_t)-\hat{f}\_{\pi\_0}(x^{(z)}\_t)\|)$ is added in *Notations and settings* in section 4.
> > >
> > > #### **Q5: Does the paper actually specify that the equation is for a special case (ideal case does to the best of my knowledge not specific to target the origin)? Please add the targets, otherwise, readers will be confused. Also, specify the pseudo-inverse symbol (dagger).**
> > >
> > > **A5:** This control expression is not scenario specific. The *“idea case”* in the article is not a proper expression and has been revised in the text. Thank you for your advice. We have specified the pseudo-inverse symbol in the text.
> > >
> > > #### **Q6: This is very confusing. You state that the lower bound $\tilde{h}(x^{(z)}\_t)>p$. exists. I am not convinced of that because of Q2 above. However, if it does, why should the same bound hold for $\pi\_\delta$?  In the proof of Lemma 1, where did $\mathbb{E}[\|w\_t\|]$ go?**
> > >
> > > **A6**  The response about Q2 is given in A2. The proof for Lemma 1 is in *Appendix A.3* and $\|\mathbb{E}\_{z\sim \pi\_\delta}(w\_t)\|=0$ because we assume that $\|\mathbb{E}(w\_t)\|=0$ in *Assumption 1*.
> > >
> > > #### **Q7: If I understand you correctly, the misspecification $\Delta B$  is unknown and Corollary 2 shows that the ACE under misspecification can be bounded (with the additional term). Please say that directly.**
> > >
> > > **A7:** Yes, it's correct.
> > >
> > >
> > > #### **Q8: It is still not understood enough about Algorithm 1.**
> > >
> > > **A8:** Thank you for pointing out this problem. $\nabla L\_{\theta}$ denotes the gradient of L w.r.t. $\theta$ so it should be $\nabla\_{\theta} L$. We have updated it to the latest version.
> > >
> > > #### **Q9: Please clarify in the table description how significant results are marked (I assume it is bold, but having checkmarks everywhere does not improve reading). Please indicate statistical significance in Tables 2 and 3!**
> > >
> > > **A9:** Thank you again for indicating this problem with the readability of tables. We have deleted the checkmarks and labelled significant results with bold digits in the latest version.

---

> ### Author Response · Authors · 2022-11-12
> **Reply to Reviewer r1g3 for previous review (Part 2)**
>
> #### **Q9: Please indicate which experimental results are statistically significant.**
>
> **A9:** Thanks for your advice, we have augmented the explanation of our experimental results in the latest revision. Following is the reply to your question.
>
> In the inverted pendulum task, it seems that we only achieve significantly lower ACE in row 3 (the gale test env. with the changed $\hat{c}$). But as the breeze task is too simple, the OoD-Control cannot show a wide gap with the OMAC. (You can see that the results of the OoD-Control, the OMAC, and even the No-adapt are close to the omniscient.) And for row 2, the strong breeze task is also not difficult enough to perform prominently better, but the OoD-Control still obtain nearly the half ACE of the OMACs.
>
> For row 4 and row 5, we aim to show that when $\hat{c}$ is unchanged, both the OMAC and the OoD-Control lose efficacy. Thanks for your valuable advice, we emphasize row 3 in the latest revision.
>
> So, why OoD-Control performs better? The typical learning method trains a control function w.r.t. the state vector from trajectories. However, when the perturbation is quite large, the state will easily run into an unseen case where the function value is missing or has a significant deviation, which causes big control error.
>
> In OoD-Control, a random noise related to the state is added to the state when training, therefore the function learns from a larger case.

---

### Official Review · Reviewer_7hFa · 2022-10-26

**Confidence:** 4
**Correctness:** 2
**Technical Novelty And Significance:** 3
**Empirical Novelty And Significance:** 2
**Recommendation:** 5

**Clarity, Quality, Novelty And Reproducibility:**

The notation and wording is tedious to follow is some instances. For example, the wording "prediction error" of the unknown dynamics is (correctly) used to refer to $ \epsilon = f(x^{(z)}_t , c) − \hat{f}(x^{(z)}_t , \hat{c}) $ (implicitly throughout the paper, explicitly page 6 lemma 2), in which case indeed "the methodology provides the guaranteed upper bound for the prediction errors of the unknown dynamics".

However, on page 4 (beginning of section 4.1) "prediction error" is used to refer to $\tilde{h}_{\pi_0}(x^{(z)}_t)$ and now ($\tilde{h}$ being monotonically decreasing) a "lower bound of the prediction error maintains constant under unpredictable disturbance".

In fact, the $\tilde{h}_{\pi_0}(x^{(z)}_t)$ shorthand is also used without explicit definition in this section. These might be small details but reduce the quality of presentation and thus reader appreciation of the core theoretical work behind the method.

The above mentioned example is not a singularity. The overall clarity of the paper, in terms of better wording and grammatical construction, could do with some improvement.

**Strength And Weaknesses:**

Strengths:
- The work deals with the very pertinent topic of OoD performance for learned dynamics
- The theoretical bound shown, without the need for the restrictive eISS assumption is a notable contribution for nonlinear under-actuated systems.
- The performance of the control protocol is impressive in simulation, not only against previous work but even with respect to omniscient ground truth controllers
- Simulation work is of good quality, permitting clean comparison of baselines and a clear presentation of the results.

Weaknesses:
The work is theoretically sound and relevant. The experimental setup however leaves some claims unexplained and many important questions unanswered.
-  The authors claim in section 5.4 (quadrotor) that "As the wind field in the testing environment is quite different from the training environment, our algorithm performs relatively better than the baseline and no-adapt." Apart from the poor grammatical phrasing, the only material in the paper that comes to support that claim is in appendix B.6.1 "The wind comes from only one octant, so it differs
from test environment in not only distribution but also direction. The models are trained in only one
environment. Our OoD-Control model adapts well in testing even though the training material is
quite little." Again neglecting the very poor wording of the last sentence, one fails to see how much OoD testing is really happening, i.e. how dissimilar the settings really are to the DNN.
- An important study would include pushing the architecture to its limits with perturbations of increasingly different distributions and directions. The out of distribution capacities claimed in the very model name OoD-Control, are not significantly exhibited.
- Along the same lines, a precise explanation of the data consumed to train the model is missing, "quite little" being an unacceptable unit of measure for a machine learning conference paper. This remark generalises to the entire training framework which remains very obscure: what model architecture is used? Of what size? What is the data pipeline setup? etc.
- What about other control methods to fight the wind ? Why only benchmark against OMAC ? The authors claim "most of the previously mentioned control
methods suffer from limitations. Imprecise system modeling and non-modeled environmental
disturbances may result in unacceptable performance or instability." It would be worth comparing the performance against state of the art robust control protocols for example to support this claim.
- The authors fail to discuss any possibility of using this algorithm in the real world. The natural limitations of data collection via Monte Carlo methods make this method trainable in simulation. What about the simulation to reality gap? Have the authors considered training a OoD-Controller in simulation and attempting to measure its performance on real hardware? Any results in this direction would be of great value.



**Summary Of The Paper:**

The work presents a novel adaptive data-driven UAV flight control method, OoD-Control. Under gaussian noise assumptions and bounded control and disturbances, the authors show that the upper bound of the prediction error remains constant under unpredictable disturbance. The control method (and bound) is also shown to be extensible to different nonlinear dynamical models.

**Summary Of The Review:**

The paper proposed a very sound construction of a novel adaptive data-driven UAV flight control method based on the derivation of a prediction error bound result. This seems to offer improved expressiveness in learning simulated wind dynamics, and thus improved performance against earlier work. The authors' claim that the architecture achieves OoD generalisation remains however questionable, or to the least has not been explored to the fullest. A lack of clarity in terms of precise training protocol as well as an absence of connection to possible real world implementations penalise the paper's impact.

---

> ### Author Response · Authors · 2022-11-12
> **Reply to Reviewer 7hFa for previous review (Part 1)**
>
> #### **Q1: How much OoD testing is really happening, i.e. how dissimilar the settings really are to the DNN?**
> **A1:** We reveal the exact settings of the two experiments as follows:
> - **Inverted Pendulum.** The wind for the inverted pendulum task has two dimensions: $x$ (horizontal) and $y$ (vertical). When training the model, we sample $x$ and $y$ from 5 different normal distributions, denoted as $\chi_k = N(0, 0.2k), k=1,2, ...,5$, respectively. Then, for the testing, we shift the distribution to the uniform distribution and set three levels: breeze, strong breeze, and gale. The details of the training sets and the test sets are shown in *Table 1*.
>
> |Inverted Pendulum|||
> |----|---|---|
> |Training Set $X$||$\chi_k = N(0, 0.2k), k=1,2,...,5$|
> |Testing Set $\Omega$|Breeze|$\omega_1^{k} = U(-0.5k, 0.5k), k = 1, 2, ..., 5, 6$|
> ||Strong Breeze|$\omega_2^{k} = U(-3.0-0.5k, 3.0+0.5k), k = 1, 2, ..., 5, 6$|
> ||Gale|$\omega_3^{k} = U(-6.0-0.5k, 6.0+0.5k), k = 1, 2, ..., 7, 8$|
>
> *Table 1: The experiment settings of the Inverted Pendulum Task.*
>
> - **Quadrotor.** The wind for the quadrotor task has three dimensions: $x$, $y$, and $z$. The distributions of the wind are similar to the distributions in the pendulum task, but some details are altered. For training, we sample wind data from only one multivariate normal distribution $N(\mu, \Sigma)$ with $\mu=[0,0,0]$ and $\Sigma=diag(1,1,1)$. Meanwhile, we make the training wind come from only one octant, which means all the training data are changed to their absolute value. So, we have $\chi = |N(\mu,\Sigma)|, \mu=[0,0,0], \Sigma=diag(1,1,1)$. For testing, we also have three levels of wind named breeze, strong breeze, and gale, but they are all multivariate uniform distributions. The details are shown in *Table 2*.
>
> |Quadrotor|||
> |----|---|---|
> |Training Set $X$||$\chi = \mid N(\mu,\Sigma)\mid, \mu=[0,0,0], \Sigma=diag(1,1,1)$|
> |Testing Set $\Omega$|Breeze|$\omega_1 = U(D_1), D_1=${$(x,y,z)\mid x,y,z \in (-3,3)$}|
> ||Strong Breeze|$\omega_2 = U(D_2), D_2=${$(x,y,z)\mid x,y,z \in (-6,6)$}|
> ||Gale|$\omega_3 = U(D_3), D_3=${$(x,y,z)\mid x,y,z \in (-8,8)$}|
>
> *Table 2: The experiment settings of the Quadrotor Task.*
>
> #### **Q2: The out-of-distribution capacities claimed in the very model name OoD-Control, are not significantly exhibited.**
>
> **A2:** To push our algorithm to the limit, we design another wind model, where the velocity of the wind is constant but the direction changes once in a while. This model is definitely stronger than the original test model which follows distribution $U(-k,k)$ when the velocity is $\sqrt3$ times greater than $k$.
>
> In this experiment, we train the models with constant speed wind ($1m/s, 2m/s$ and $3m/s$), and test them with still constant speed wind ($4m/s, 8m/s, 12m/s$ and $16m/s$). We judge whether a model is capable of tracking for a long period since an incompetent policy will cause exponential loss over time. Table 3 shows the average loss when the testing period is 60 seconds.
>
> The loss of the omniscient model under no-wind condition (0.0395) is recorded as the inherent tracking error of mechanical systems. And we use five times the loss as the threshold to determine whether the model is under control.  Under this premise, OoD-Control is capable of tracking under constant $12m/s$ wind.
>
> ||$4m/s$|$8m/s$|$12m/s$|$16m/s$|
> |--|--|--|--|--|
> |OMAC|0.30916 (0.32181)|0.42912 (0.44565)|0.50627 (0.53069)|0.64843 (0.66788)|
> |No-adapt|0.34119 (0.35692)|0.44982 (0.46950)|0.56924 (0.58756)|0.61112 (0.64046)|
> |**OoD-Control**|**0.10049 (0.11943)**|**0.12969 (0.16163)**|**0.16560 (0.19650)**|**0.31772 (0.47589)**|
>
> *Table 3: The experiment results of the test on the wind with constant speed (pushing the model to its limits).*

---

> ### Author Response · Authors · 2022-11-12
> **Reply to Reviewer 7hFa for previous review (Part 2)**
>
> #### **Q3: What model architecture is used? Of what size? What is the data pipeline setup? etc.**
>
> **A3:** Our DNN model for representation learning has three linear layers with spectral normalization. And we use ReLU as the activation function for the outputs of the first and the second layers. We give the details of our model architecture in Table 4. Actually, we set our model architecture consistent with OMAC for a fair comparison of the methodology itself.
>
> In the inverted pendulum task, we have 5 training iterations and 6 or 8 testing iterations (6 for breeze and 8 for strong breeze). Each iteration's duration is 40 seconds.
>
> In the quadrotor task, we have only 1 training iteration and 1 testing iteration for each wind condition. Each iteration's duration is 10 seconds. Additionally, we run the training and testing iterations with 10 different random seeds.
>
> |**DNN Architecture**|||
> |--|--|--|
> |**Inverted Pendulum**|||
> |**Layer Type**|**Size**|**Activation Function**|
> |SN (linear)|(2, 25)|ReLU|
> |SN (linear)|(25, 30)|ReLU|
> |SN (linear)|(30, 20)|None|
> |**Quadrotor**|||
> |**Layer Type**|**Size**|**Activation Function**|
> |SN (linear)|(13, 50)|ReLU|
> |SN (linear)|(50, 100)|ReLU|
> |SN (linear)|(100, 33)|None|
>
> *Table 4: The DNN architecture used for the two experiment tasks.*
>
> #### **Q4: What about other control methods to fight the wind? It would be worth comparing the performance against state-of-the-art robust control protocols for example to support this claim.**
>
> **A4:** We compare our method with the reinforcement learning method *GIE-CLF*[1]. This algorithm adds a random control term to the model in training to improve robustness. As shown in Table 5, under the same training duration, OoD-Control achieves 20% better performance than the *GIE-CLF* method.
>
> |Training & Testing Duration|Training & Testing Epoch|Test Env.|GIE-CLF|**OoD-Control**|
> |--|--|--|--|--|
> |40|5|$U(-8,8)$|0.206(0.045)|0.166(0.036)|
> |40|5|$U(-10,10)$|0.288(0.053)|0.238(0.049)|
>
> #### **Q5: Correct the notation and wording mentioned in "Clarity, Quality, Novelty And Reproducibility".**
>
> **A5:** In this paper, the prediction error of the unknown dynamics is referred to $\epsilon={f}(x^{(z)}_t,{c})-\hat{f}(x^{(z)}_t,\hat{c})$. We add an explicit definition of $\tilde{h}(x_t^{(z)})$ in the revised version before section 4.1.
>
> **References:**
>
> [1] Anonymous. (2022). *Learning Control Lyapunov Functions For
> High-Dimensional Unknown Systems Using
> Guided Iterative State Space Exploration.* https://openreview.net/forum?id=YHxp8eRry6F

---

### Author Response · Authors · 2022-11-12
**Reply to all reviewers for common questions**

Thank you very much for your precious and constructive advice. We have observed your concern about applying the method to the real world. For example, *Reviewer 7hFa* pointed out that we "fail to discuss any possibility of using this algorithm in the real world", and *Reviewer ma1x* indicated that our method "is not evaluated on a real system to convince that the proposed method can be applied to a real UAV."

We have set this assignment in our future work. Now the designed real UAV is ready, and we are testing the control algorithms. The objective of this paper is the theoretical derivation and implementation of our method. Though validating our method in the real world is not the priority of this work, we still thank the reviewers for pointing out this problem.

---

### Decision · Program_Chairs · 2023-01-20

**Decision:**

Reject

**Justification For Why Not Higher Score:**

Please see weaknesses.

**Justification For Why Not Lower Score:**

N/A

**Metareview: Summary, Strengths And Weaknesses:**

The paper describes a method for adapting the control of unmanned aerial vehicles (UAVs) to unknown environments with different data distributions from the training set (the OOD problem). The OoD-Control method is based on a theoretical bound that guarantees a constant upper bound on the control error within a certain range of perturbations on the states. The authors claim that the method performs state-of-the-art in positioning stability and trajectory tracking problems on UAV dynamic models.

Strengths:
- The theoretical bound shown, without the need for the restrictive eISS assumption is a notable contribution to nonlinear under-actuated systems.
- The performance of the control protocol is impressive in simulation.
- The problem is very relevant

Weaknesses:
- Reviewers raised concerns about the applicability of the model to real-world scenarios. The authors acknowledged this shortcoming and referred it to future work. However, I also believe that given the nature of the work and the proposed method, its real impact could be justified with real-world deployment.

- Although the theoretical concerns were addressed during the rebuttal, I believe that a significant revision of the paper, which more clearly explains the main argument and how each theoretical statement fits into it, is necessary before accepting the paper.

I strongly advise the authors to take the reviewer's comments and improve the paper with real-world experiments and a major rewriting of the contributions in a way that concisely structure the theoretical analysis in order to avoid future confusions.

**Summary Of Ac-Reviewer Meeting:**

N/A